# The intensities of canonical senescence biomarkers integrate the duration of cell-cycle withdrawal

Humza M. Ashraf[1,2], Brianna Fernandez[1,2] & Sabrina L. Spencer ●[1,2] ✉

Senescence, a state of irreversible cell-cycle withdrawal, is difficult to distinguish from quiescence, a state of reversible cell-cycle withdrawal. This difficulty arises because quiescent and senescent cells are defined by overlapping biomarkers, raising the question of whether these states are truly distinct. To address this, we use single-cell time-lapse imaging to distinguish slow-cycling cells that spend long periods in quiescence from cells that never cycle after recovery from senescence-inducing treatments, followed by staining for various senescence biomarkers. We find that the staining intensity of multiple senescence biomarkers is graded rather than binary and reflects the duration of cell-cycle withdrawal, rather than senescence per se. Together, our data show that quiescent and apparent senescent cells are nearly molecularly indistinguishable from each other at a snapshot in time. This suggests that cell-cycle withdrawal itself is graded rather than binary, where the intensities of senescence biomarkers integrate the duration of past cell-cycle withdrawal.

Senescence is a state of irreversible cell-cycle withdrawal associated with aging and DNA damage. Extended durations of recovery from DNA damaging treatments lead to cell-cycle re-entry and population regrowth[1,2], but it is unknown whether this regrowth phenotype is caused by cells that re-enter the cell cycle from a reversible state of arrest called quiescence or whether it is the result of a proliferative subpopulation that outcompetes senescent cells over time (Fig. 1a). It is challenging to study reversible vs. irreversible cell-cycle withdrawal since these fates are nearly indistinguishable from each other at a single point in time, making it unclear which cells will go on to cycle in the future vs. which cells will remain arrested[3]. These limitations have led to speculation that some cells can escape from senescence to re-enter the cell cycle[2,4], but it has not been shown that these cells were truly senescent to begin with. As a result, there is a critical need to accurately detect senescent cells to clarify whether quiescence and senescence are binary, distinct cellular states or whether they exist on a gradient of cell-cycle withdrawal.

The gold-standard marker for detecting senescent cells is the colorimetric senescence-associated beta-galactosidase (SA-β-Gal) stain[3,5]. However, the β-galactosidase gene is dispensable for the induction and maintenance of senescence[6], raising questions about a causal relationship between SA-β-Gal positivity and irreversible cell-cycle withdrawal. Furthermore, the colorimetric nature of the standard SA-β-Gal stain makes it challenging to quantify. While more quantitative fluorescent senescence-detection kits now exist, most studies still classify cells by simply binarizing the classic colorimetric SA-β-Gal stain by manually labelling cells either blue (senescent) or not blue (not senescent). Due to these limitations, studies often measure additional senescence markers in separate parallel experiments. These include the lack of cell cycling (e.g. Ki67 or phospho-Rb), expression of Cyclin-Dependent Kinase (CDK) inhibitors (e.g. p21 or p16), DNA damage (e.g. 53BP1 or γH2AX), presence of the senescence-associated secretory phenotype (SASP, with IL6 and IL8 being among the most common factors), loss of Lamin B1 (a structural component of the nuclear lamina[7]), and increased cell size[3,8]. Since no single senescence marker is unique to senescence, multiplexing multiple markers in single cells has recently been explored to identify senescent cells more accurately[3,8,9]. However, no study has systematically tested these markers to quantify their predictive power for identifying senescent cells.

[1]Department of Biochemistry, University of Colorado, Boulder, CO 80303, USA. [2]BioFrontiers Institute, University of Colorado, Boulder, CO 80303, USA.
✉ e-mail: sabrina.spencer@colorado.edu

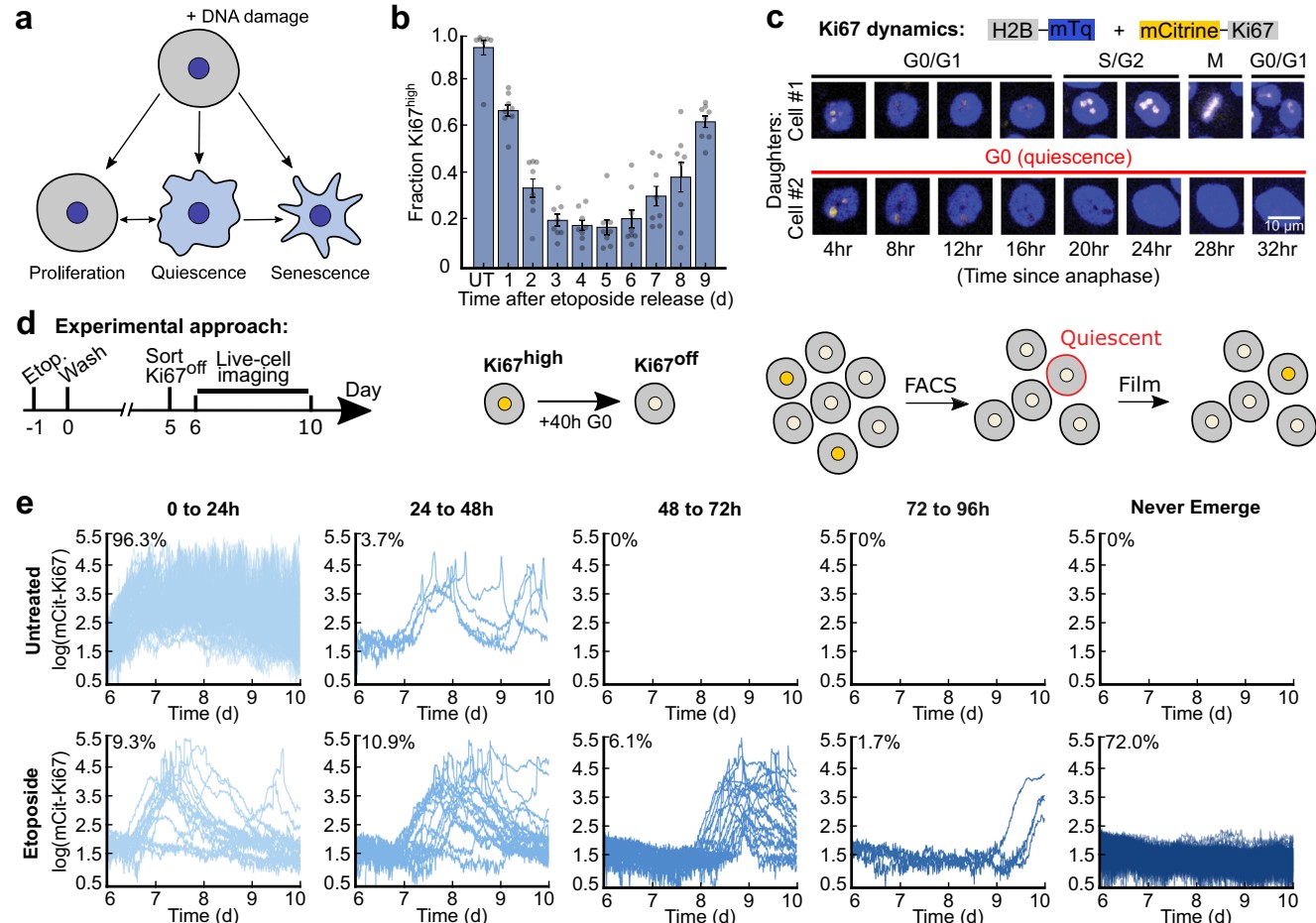

**Fig. 1 | A subpopulation of cells exits quiescence to re-enter the cell cycle after DNA damage stress. a** Multiple cell fates arise during recovery from acute DNA damaging agents. **b** MCF10A cells were treated with 10 μM etoposide for 24 h before being washed and allowed to recover for 1 to 9 d. Cells were fixed and stained for Ki67, and the fraction of Ki67off cells was calculated for each condition. UT, untreated. **c** Dynamics of mCitrine-Ki67 with respect to cell-cycle phase in two daughter cells originating from the same mother. The top daughter proceeds through the cell cycle, while the other daughter enters a prolonged quiescence.

**d** Experimental schematic: MCF10A cells expressing endogenously tagged mCitrine-Ki67 were treated on day -1 with 10 μM etoposide for 24 h and washed on day 0. On day 5, Ki67off cells were isolated by flow cytometry, plated, and allowed to grow for 24 h before being imaged for 96 h by time-lapse microscopy. **e** Single-cell traces are grouped based on their relative timing of cell-cycle re-entry from the Ki67off state; the percentage of cells in each group is indicated. 200 cell traces total are plotted in each row.

## Results

### A subset of cells re-enters the cell cycle from quiescence after etoposide treatment

To measure the heterogeneity in cell-cycle fates following senescence induction, we treated MCF10A non-transformed mammary epithelial cells with etoposide, a commonly used anti-cancer chemotherapeutic agent that inhibits topoisomerases to induce DNA damage and senescence[10]. MCF10A cells were released for 1–9 d from a 24 h treatment of 10 μM etoposide, and cells were fixed and stained for the proliferation marker Ki67 (Fig. 1b). While the majority of cells initially withdrew from the cell cycle, the population began to rebound at day 6 of drug recovery. This proliferation rebound was confirmed in MCF10A cells with an alternative proliferation marker (Rb phosphorylation, a cell-cycle marker that turns on once cells commit to the cell cycle and turns off when cells exit the cell cycle)[11], as well as in RPE-hTERT, MCF7, and WI38-hTERT cells (Supplementary Fig. 1a–d). Furthermore, while we observed decreases in the proportion of proliferating cells at increasing doses of etoposide, there was no concentration of drug (up to 50 μM) that eliminated all cycling cells to yield a pure senescent population (Supplementary Fig. 1d).

The proliferating cells rapidly overtake the non-cycling cells by day 9 after etoposide treatment, but it remains unclear what proportion of the non-cycling cells at a snapshot in time will re-enter the cell cycle in the future. To address this question, we used MCF10A cells in which Ki67 was tagged at the endogenous locus with mCitrine[12] (Fig. 1c) to isolate non-cycling Ki67off cells by flow cytometry 5 d after etoposide release (Fig. 1d), as this was the time window with the fewest cycling cells (Fig. 1b). The levels of Ki67 protein decay with second order kinetics upon cell-cycle exit, hitting the floor of detection after

Here, we used long-term single-cell time-lapse imaging of cell-cycle reporters to classify cells as fast-cycling, slow-cycling, or predicted-senescent during the course of a 4-day movie after extended recovery from acute DNA damage. We mapped these cell-cycle behaviors to *post hoc* SA-β-Gal staining by developing a method for quantifying and multiplexing the stain with other senescence biomarkers. We found that the relative blueness of the SA-β-Gal stain reflects increased lysosomal content and scales with increasing durations of cell-cycle withdrawal, rather than with senescence per se. Furthermore, all other senescence markers tested also scaled with the duration of cell-cycle withdrawal, including LAMP1, cell size, IL8, 53BP1, p21, and Lamin B1. We conclude that the relative intensities of canonical senescence biomarkers integrate the duration of cell-cycle withdrawal, rather than being unique to senescence.

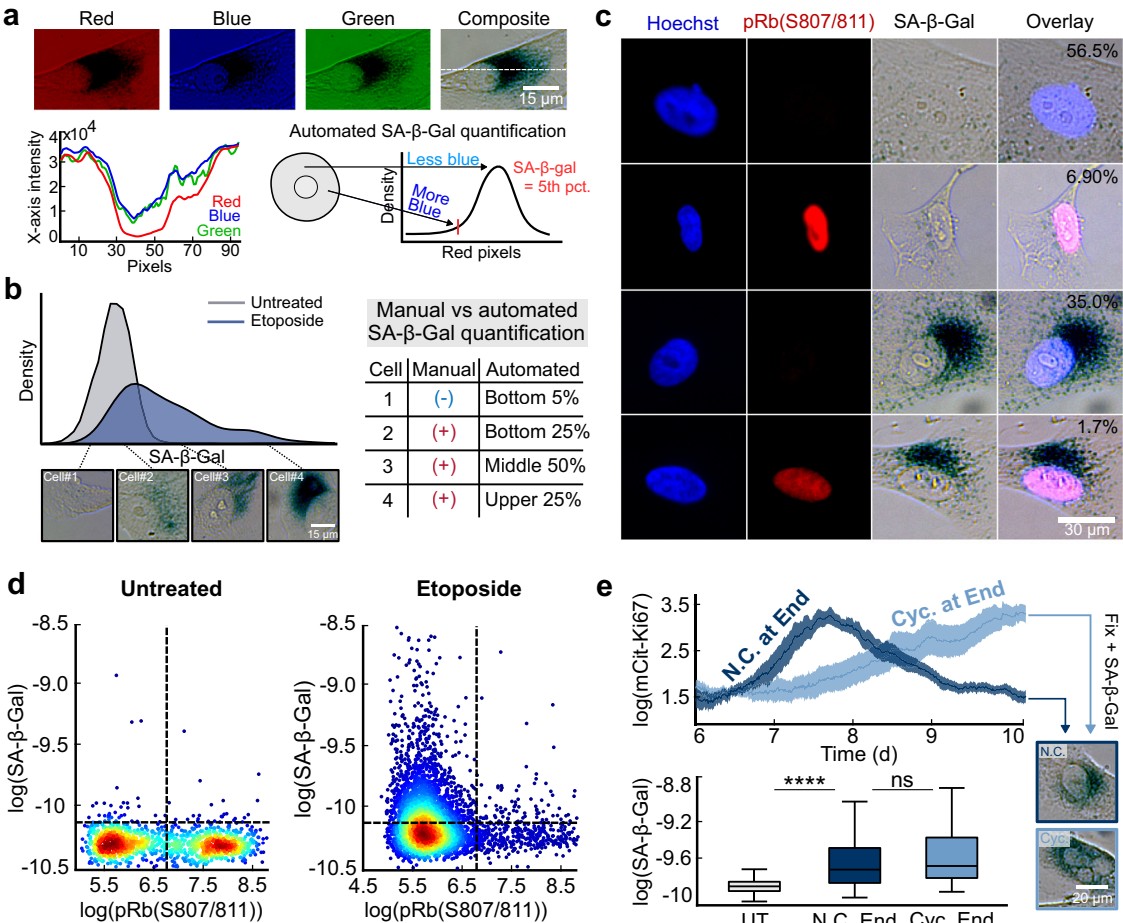

**Fig. 2 | Quantifying SA-β-Gal in single cells reveals a graded signal with overlap between treated and untreated cells and the presence of SA-β-Gal^pos cycling cells. a** A representative single MCF10A cell stained for SA-β-Gal and imaged in pseudo-color-brightfield at 6 d after release from a 24 h treatment with 10 μM etoposide. An intensity profile (dotted line) was taken from each channel of the RGB stack. We define the SA-β-Gal signal as the 5th percentile of the red pixels in the cytoplasm of each cell. **b** Distribution of SA-β-Gal signal in untreated cells or in cells released for 6 days after release from a 24 h treatment with 10 μM etoposide, with four representative single cells at increasing intensities of staining. **c** Heterogeneity in co-staining of SA-β-Gal and Rb phosphorylation by immunofluorescence in

MCF10A cells released for 6 d from a 24 h treatment with 10 μM etoposide. Percentages reflect the fraction of cells with each behavior using the cutoffs in **d**. **d** Scatter plots of SA-β-Gal versus phospho-Rb from the same cells in **c**. Heatmap represents the density of data points. **e** The same data from Fig. 1d for etoposide-released cells that entered the cell cycle during live-cell imaging. Cells were split into those that completed a cell cycle during the 4 days of imaging but were Ki67^off again by the final frame of the movie (N.C. at End) vs. those that were in the cell cycle and Ki67^high on the final frame of the movie (Cyc. at End). Cells were fixed and stained for SA-β-Gal after the last frame of the movie was taken and each cell's SA-β-Gal signal was linked to its Ki67 history.

40 h[12]. Thus, Ki67^off cells at the time of sorting had been out of the cell cycle for 40 h or more. Immediately after sorting, we replated the Ki67^off cells and began filming them the following day for an additional 4 days. 100% of the untreated, rare spontaneously quiescent Ki67^off cells re-entered the cell cycle within the first 2 days of filming, consistent with their quiescent status at the time of sorting. Surprisingly, despite the strong senescence-inducing conditions, 28% of etoposide-released cells resumed proliferation at some point during live-cell imaging (Fig. 1e). Similar results were obtained when this experiment was repeated with 10 Gy ionizing radiation (Supplementary Fig. 1e, f). Thus, a significant fraction of non-cycling cells 5 d after acute DNA damage are not truly senescent, since they are fated to re-enter the cell cycle in the future.

## Quantification of SA-β-Gal reveals a gradient in staining and overlap with cycling cells

Having developed a flow cytometry and time-lapse approach to classify cells by their cell-cycle status, we sought to clarify the relationship between cell-cycle withdrawal and SA-β-Gal staining, the gold-standard marker of senescence. To address the critical need in the senescence

field for quantification of the SA-β-Gal stain, we adapted an existing method[13] to develop an automated, high-throughput strategy for measuring SA-β-Gal in thousands of single cells (see Methods). We used the red component of the red-green-blue (RGB) image of the stain to compute a single value from the distribution of pixels within the cytoplasm of each segmented cell. We chose the red channel because the SA-β-Gal stain is primarily composed of blue and green pigments that preferentially absorb red light. Thus, the SA-β-Gal stain can be most easily quantified as the absence of red signal within every cell, and this channel has the largest dynamic range relative to background (Fig. 2a). We used the value at the 5th percentile of red pixels within the cytoplasm of each cell as the SA-β-Gal score (Supplementary Fig. 2a, b), since this method visually matched the relative blueness of cells (Supplementary Fig. 3a) and recapitulated the gradient of staining in single cells induced to senescence (Fig. 2a, b). Even though SA-β-Gal is almost always classified manually as a binary marker of senescence (blue or not blue), we found that there is actually no clear cutoff for designating a cell as senescent due to the gradient of blueness (Fig. 2b).

Next, we co-stained cells for SA-β-Gal and phosphorylated Rb, a marker of cell cycle commitment[11], and discovered a surprisingly

heterogeneous mixture of behaviors. Because the SA-β-Gal signal is graded (Fig. 2b), we initially used a cutoff at the 95th percentile of untreated cells to designate a cell as SA-β-Gal-positive (hereafter SA-β-Gal[pos]), since these cells are "bluer" than baseline. Although most SA-β-Gal[pos] cells were phospho-Rb[low], consistent with what would be expected for senescent cells, we also identified SA-β-Gal[neg]/phospho-Rb[high] cycling cells and SA-β-Gal[neg]/phospho-Rb[low] presumably quiescent cells. Surprisingly, we also identified a small fraction (1.7%) of SA-β-Gal[pos]/phospho-Rb[high] cells. This latter population calls into question the reliability of SA-β-Gal as a senescence marker, since no truly senescent cell should ever be in the cell cycle (Fig. 2c). However, comparing the relative intensities of SA-β-Gal staining following etoposide release revealed that the bluest cells in the population were significantly more likely to be phospho-Rb[low] compared to less-blue cells, which were associated with more variability in phospho-Rb status (Fig. 2d). This suggests that the confidence in classifying cells as senescent increases as a function of the intensity of SA-β-Gal staining, with intermediate levels of SA-β-Gal staining encompassing both potentially reversibly and irreversibly arrested cells.

To determine the origin of the SA-β-Gal[pos]/phospho-Rb[high] cells, we returned to our data set from Fig. 1e where the cells were also stained for SA-β-Gal at the end of the movie. Because SA-β-Gal[pos]/phospho-Rb[high] cells tended to have intermediate levels of SA-β-Gal staining, we hypothesized that this subpopulation might represent slow-cycling cells that we showed in Fig. 1e to be easily misclassified as senescent. To test this, we split the slow-cycling etoposide-released subpopulation that re-entered the cell cycle during imaging into two categories: slow-cycling cells that happened to be in the cell cycle at the final frame of the movie, and cells that cycled earlier in the movie but were not in the cell cycle at the final frame of the movie (Fig. 2e, top). We found that slow-cycling cells that happened to be in the cell cycle at the final frame of the movie have significantly higher levels of SA-β-Gal staining compared to untreated control cells. This explains the origin of the SA-β-Gal[pos]/phospho-Rb[high] subpopulation (Fig. 2e, bottom) as cells that were withdrawn from the cell cycle for a long period and just recently re-entered the cell cycle. However, we detected no significant difference in the relative levels of blueness of slow-cycling cells that were in the cell cycle vs. were not in the cell cycle at the final frame of the movie. This suggests that 1) cell cycle re-entry does not immediately extinguish the SA-β-Gal signal, and that longer periods of proliferation may be required to fully eliminate SA-β-Gal-positivity (Fig. 2e, bottom), and 2) quiescent cells that are capable of cycling in the future also become SA-β-Gal[pos].

### SA-β-Gal scales with increased durations of cell-cycle withdrawal

Since slow-cycling cells also stained positive for SA-β-Gal, we questioned whether the duration of cell-cycle withdrawal could account for the heterogeneity in staining. To test this, we classified cells by their cell-cycle status by filming a live-cell sensor for CDK2 activity[14] from days 6–10 following etoposide release. CDK2 activity begins to rise when cells commit to the cell cycle and increases steadily thereafter until mitosis. By contrast, cells turn off CDK2 activity and enter a CDK2[low] state when they exit the cell cycle[14] (Fig. 3a). At the end of the time-lapse imaging on day 10, we fixed and stained the cells for SA-β-Gal and mapped each cell's stain to its cell-cycle history over the previous 4 days (Fig. 3b). Binning cells into the top, middle, and bottom 10% of SA-β-Gal signal revealed that the intensity of staining was proportional to the total duration of time that cells spent out of the cell cycle in a CDK2[low] state, suggesting that SA-β-Gal staining scales with increasing durations of cell-cycle withdrawal (Fig. 3c, left).

Next, we tested whether SA-β-Gal-positivity could resolve fast-cycling from slow-cycling cells that exist in untreated populations experiencing natural endogenous stresses[14,15]. To test this, we sorted the bottom 1% of mCitrine-Ki67 signal by flow cytometry to enrich for the intrinsically slow-cycling cells in the population (Fig. 3c right). The

cells were re-plated after sorting and their CDK2 activities were filmed over the subsequent 2 days. We classified cells as either SA-β-Gal[pos] or SA-β-Gal[neg] by staining them immediately after filming and found that cells designated SA-β-Gal[pos] spent significantly more hours in the CDK2[low] state compared to SA-β-Gal[neg] cells. This finding supports the notion that SA-β-Gal is a general readout of increased durations of cell-cycle withdrawal even in untreated cells that are not senescent, since all untreated Ki67[off] cells re-enter the cell cycle (Fig. 1e).

To investigate the generalizability of our findings across different cell-cycle withdrawal mechanisms, we examined the median SA-β-Gal signal in cells forced into quiescence by four well-established methods: contact inhibition, serum starvation, CDK4/6 inhibition (Palbociclib), and Mek inhibition (Trametinib) for 3, 6, 9, and 12 days (Fig. 3d)[16]. Our results show that the SA-β-Gal signal increased with the duration of treatment in the contact-inhibited, Palbociclib, and Trametinib quiescence conditions (Fig. 3e and Supplementary Fig. 4a). This finding is consistent with previous observations that elevated SA-β-Gal activity is not necessarily unique to irreversible cell-cycle arrest[17,18]. While SA-β-Gal staining correlates with the time spent out of the cell cycle for several senescence and quiescence-inducing treatments (Fig. 3f), serum-starved cells may be an exception since these cells accumulate significantly less SA-β-Gal compared to the other treatments. This finding is consistent with the notion that senescence arises from cells simultaneously experiencing pro-proliferative signals (here via signaling from growth factors in the media) and unscheduled anti-proliferative signals (here due to space constraints or drug treatment)[3,8,19]. By contrast, low serum more closely resembles the natural state of quiescence in the body and therefore may create less of a clash.

Importantly, the cell-cycle withdrawal induced by contact inhibition, serum starvation, and Trametinib treatments was not driven by increases in cell size and was reversible after a 5-day release from 2 weeks of treatment, with >98% of the cells re-entering the cell cycle (Fig. 3e and Supplementary Fig. 4b). The one exception was Palbociclib, where approximately 40% of cells failed to re-enter the cell cycle, and cell size increased over time (Fig. 3e and Supplementary Fig. 4b, c). This suggests that some Palbociclib-treated cells may transition from reversible to irreversible arrest with extended treatment. Together, these data show that the heterogeneity in SA-β-Gal staining is reflective of biological heterogeneity, where cells that cycle less often under stress accumulate more SA-β-Gal staining over time.

### Increased SA-β-Gal staining reflects increased lysosomal content and autophagy

Why is SA-β-Gal staining is so closely coupled with cell-cycle status when the enzyme itself is dispensable for the induction and maintenance of senescence[6,20,21]? Since the β-galactosidase enzyme is localized to the lysosomes, we reasoned that increased SA-β-Gal staining could simply be a readout of increased lysosomal content, as has been previously reported but never displayed in single cells[20,21]. To test this, we multiplexed measurements of SA-β-Gal and LAMP1, a membrane-embedded lysosomal protein[22], and found that they co-localized and that the levels of both simultaneously increased following release from etoposide (Fig. 3g). These data reinforce the long-standing notion that increased SA-β-Gal staining following cell stress is a consequence of increased lysosome biogenesis, which has previously been shown to drive increased expression of β-Gal protein[20,21].

We next questioned whether the increased lysosome content following etoposide release was associated with changes in autophagy, since previous literature has suggested that senescent cells undergo increased autophagic flux to manage the accumulation of cellular damage and mount the SASP[23]. To investigate this idea, we measured the autophagic flux in untreated and etoposide-released cells by comparing the relative increase in LC3II protein, a commonly used marker of autophagosome formation[24,25], following a 3 h treatment of

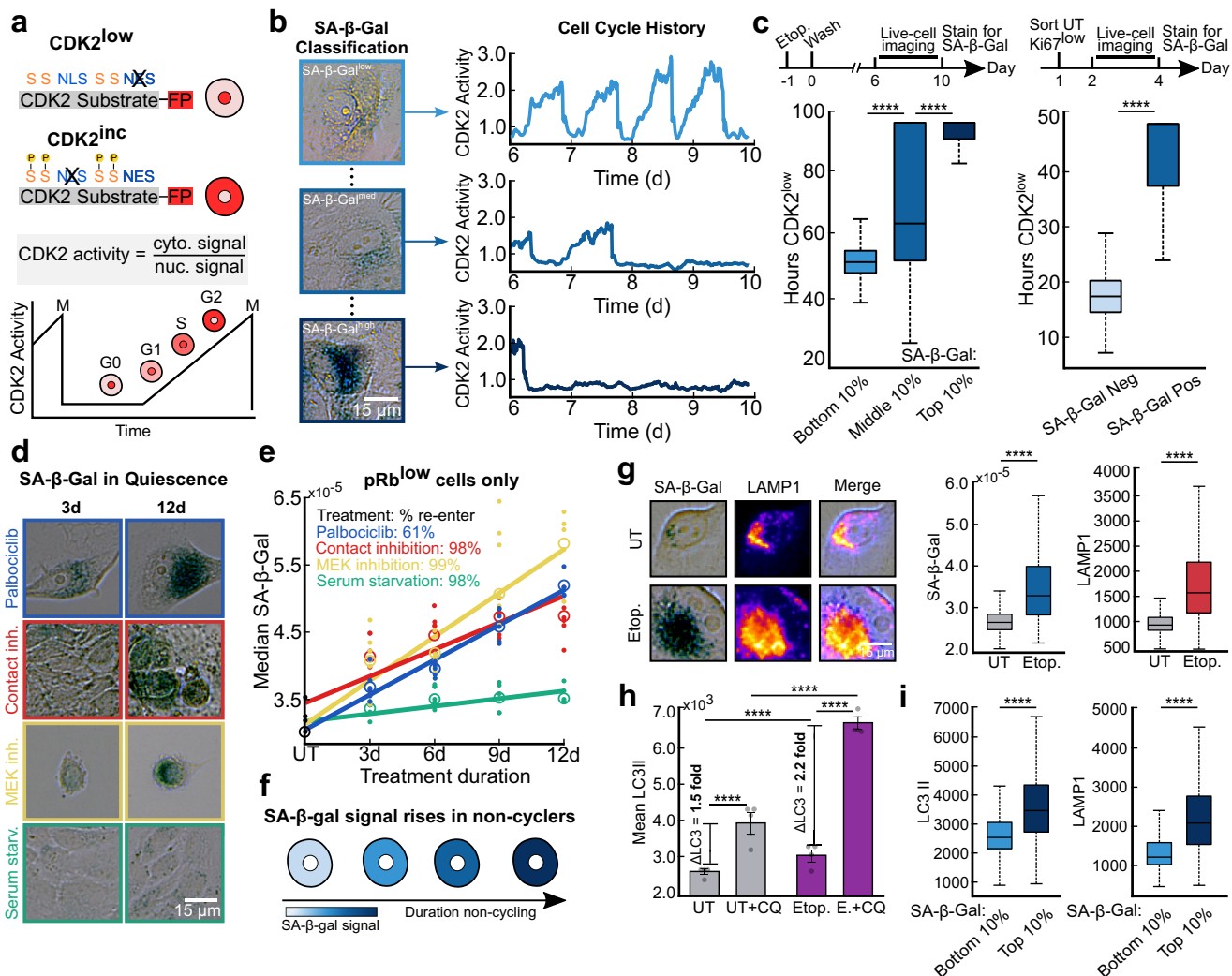

**Fig. 3 | SA-β-Gal staining marks long durations of cell-cycle withdrawal and is correlated with increased lysosomal content and autophagic flux. a** Schematic of the CDK2 activity sensor. The sensor localizes to the nucleus when unphosphorylated; progressive phosphorylation by CDK2 leads to translocation of the sensor to the cytoplasm. NLS, nuclear localization signal; NES, nuclear export signal; S, CDK consensus phosphorylation sites on serine. **b** MCF10A cells expressing the CDK2 activity sensor were treated with 10 μM etoposide for 24 h, washed, and subjected to time-lapse microscopy of CDK2 activity 6 d later for 96 h (from 6-10 d). The cells were fixed and stained for SA-β-Gal after the last frame was taken. **c** Left: Single-cell traces were clustered based on the top, middle, and bottom 10% of SA-β-Gal signal and the total hours CDK2$^{low}$ (below a cutoff of 0.8) was plotted for each bin. **c** Right: Untreated MCF10A cells were sorted by flow cytometry for the bottom 1% of mCitrine-Ki67 signal, plated and allowed to grow for 48 h, filmed for 48 h to monitor CDK2 activity, fixed and stained for SA-β-Gal, and were manually classified as SA-β-Gal positive versus negative. **d, e** Median SA-β-Gal signal for pRb$^{low}$ MCF10A cells pushed into quiescence by contact inhibition, serum starvation, 3 μM Palbociclib treatment, or 100 nM Trametinib treatment for 3-12 d, and fixed and stained for SA-β-Gal and phospho-Rb. Best fit lines were computed for each condition from the average of 6 technical replicates. Dots represent raw data points and open circles represent the mean of these data points. Percentages listed represent the proportion of cells that re-entered the cell cycle during a 5 d release from 2 weeks of treatment (see Supplementary Fig. 4c). **f** Model for SA-β-Gal accumulation as a function of cell-cycle exit time. **g** MCF10A cells were treated with 10 μM etoposide for 24 h, washed, fixed after 3 d, and stained for SA-β-Gal and LAMP1. **h, i** Same experimental scheme as described in **g**. Cells were fixed and stained for SA-β-Gal, LC3II, and LAMP1 after a 3 h treatment with 50 μM chloroquine.

50 μM chloroquine (CQ), a lysosomotropic agent that impairs autophagosome fusion[25,26]. Etoposide-released cells experienced a 2.2-fold increase in average LC3II protein levels following CQ treatment compared to control cells, which had a 1.5-fold increase. This result suggests that autophagy is significantly upregulated in cells released from etoposide (Fig. 3h).

Finally, to test whether increased SA-β-Gal staining is correlated with both increased lysosomal content and/or autophagic flux, we co-stained cells for SA-β-Gal and either LAMP1 or LC3II following etoposide release. As expected, the levels of both proteins were significantly higher in cells with the highest levels of SA-β-Gal compared to cells with the lowest levels of SA-β-Gal (Fig. 3i). Thus, SA-β-Gal staining reflects increased lysosomal content, which reflects increased autophagic flux in cells induced to senescence.

## Canonical senescence biomarkers integrate the duration of cell-cycle withdrawal

Because SA-β-Gal staining scaled with increased durations of cell-cycle withdrawal, we next asked whether other markers of senescence follow the same trend. To test this, we measured SA-β-Gal, LAMP1, cytoplasmic area, nuclear area, IL8 protein, 53BP1 (a protein that forms foci at sites of DNA damage[27]), p21, and Lamin B1 immediately after a movie that spanned days 6–10 after release from a 24 h treatment with 10 μM etoposide. First, we classified cells as either fast-cycling, slow-cycling, or predicted-senescent based on the number of hours spent in the CDK2$^{low}$ state (Fig. 4a, b and Supplementary Fig. 5a), with predicted-senescent cells being defined as those that were CDK2$^{low}$ for the entire movie (Supplementary Movie 1). For the first 7 markers, the intensity of staining was highest for predicted-senescent cells, intermediate for

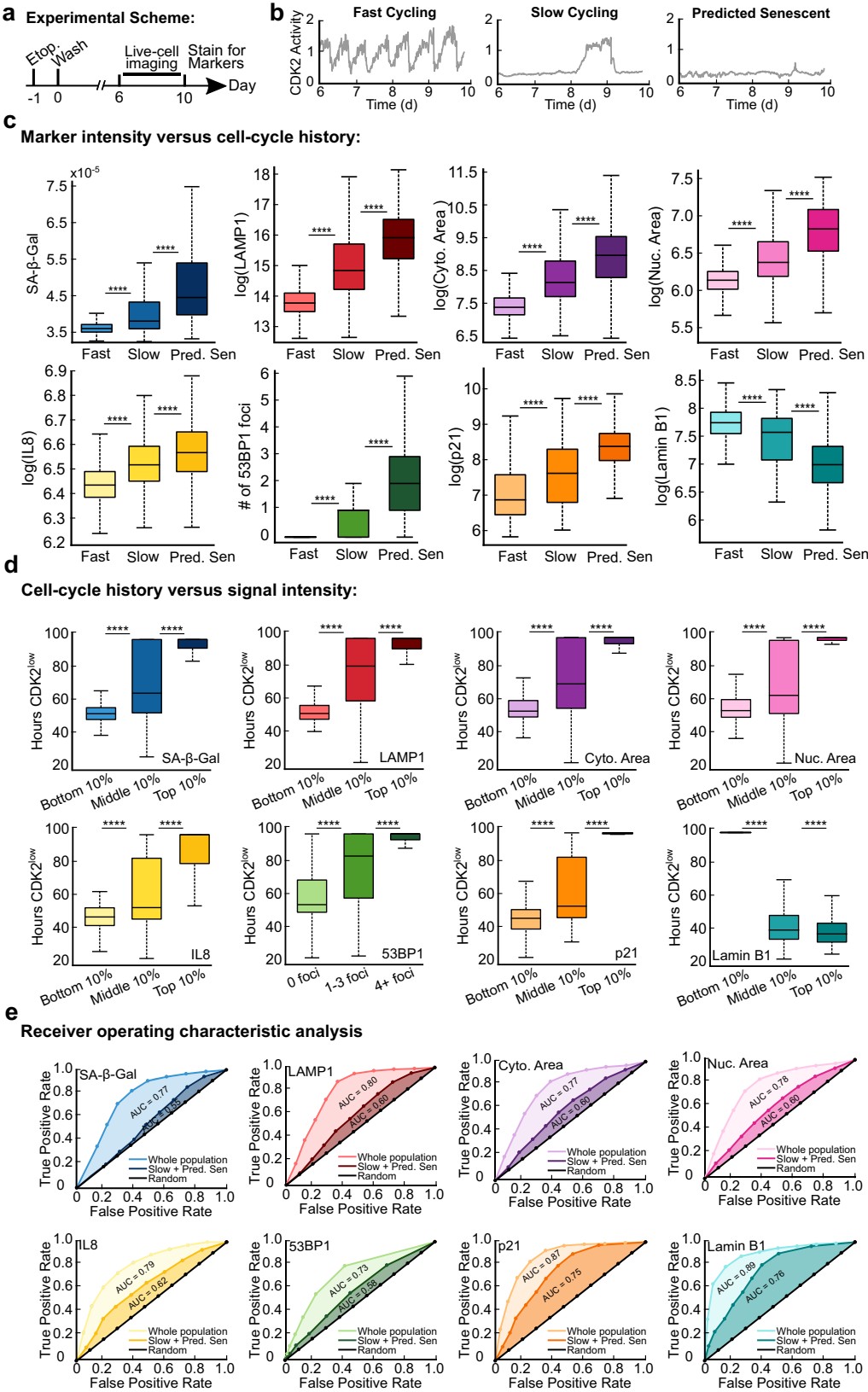

slow-cycling cells, and lowest for fast-cycling cells, while Lamin B1 followed the opposite trend (Fig. 4c, Supplementary Fig. 6, and Supplementary Movie 1).

Second, we grouped cells based on marker staining intensity into the top, middle, and bottom 10% and plotted the time the cells had spent in the $CDK2^{low}$ state over the prior 4 days. For each marker, a graded trend was observed, where the intensity of staining of the marker was correlated with the duration withdrawn from the cell cycle, with Lamin B1 again following the opposite trend compared to the others (Fig. 4d). These data suggest that the relative intensities of

**Fig. 4 | Senescence biomarker intensities reflect cell-cycle histories and can resolve predicted-senescent cells with varying levels of accuracy. a-e** MCF10A cells expressing the CDK2 activity sensor were treated with 10 μM etoposide for 24 h, washed, and subjected to time-lapse microscopy from 6-10 d after release. The cells were fixed and stained for SA-β-Gal, LAMP1, succinimidyl ester, Hoechst, IL8, 53BP1, p21, and Lamin B1 after the last frame was taken. **b, c** Cells were split into fast cycling, slow cycling, or predicted senescent based on their duration spent

CDK2$^{low}$ during live cell imaging, and the intensity of each marker was plotted for each cellular behavior. **d** The duration cells spent in the CDK2$^{low}$ state was plotted against the bottom, middle, and top 10% of signal for each marker. **e** ROC analysis using markers to identify predicted-senescent cells amidst all cells (fast-cycling, slow-cycling, and predicted-senescent cells), or amidst only slow-cycling and predicted-senescent cells. AUC indicates the area under the curve for each condition.

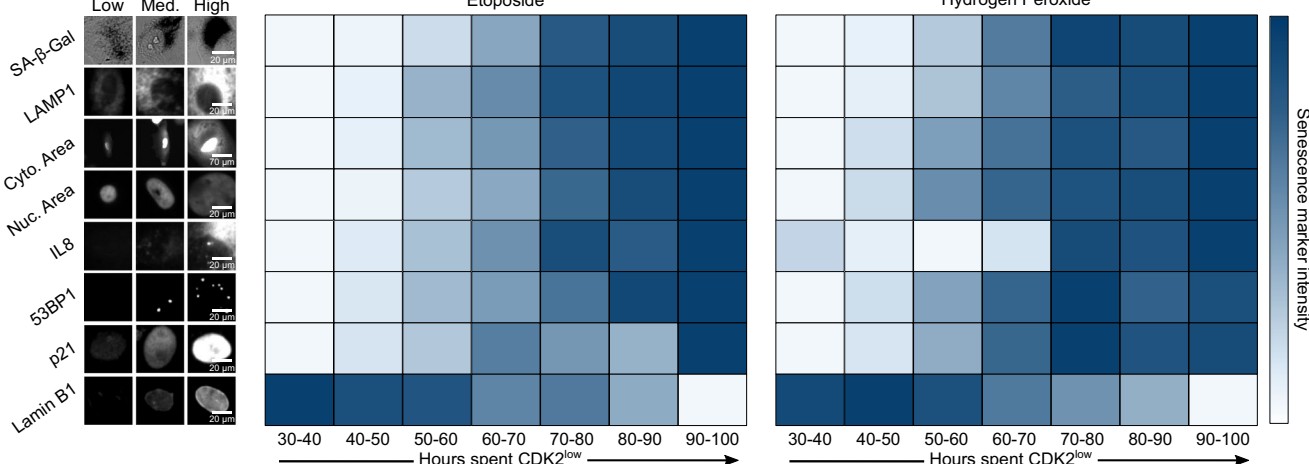

**Fig. 5 | Cell-cycle withdrawal is graded rather than binary.** Cells were grouped based on the amount of time spent in the CDK2$^{low}$ state (see Supplementary Fig. 7b). The Etoposide condition is data from Fig. 4. For the hydrogen peroxide condition, cells were treated for 2 h with 100 μM H$_2$O$_2$ in OptiMEM before being

washed twice and immediately filmed for 4d. The marker intensities in each group (see Supplementary Fig. 7a) were averaged, scaled between 0 and 1, and each box was colored accordingly. This reveals a gradual, monotonic change in each marker signal as a function of the duration of cell-cycle withdrawal.

canonical senescence biomarkers encode, in a snapshot, information about the proliferative histories of single cells released from acute DNA damage.

To quantitatively compare the powers of these senescence markers to accurately identify predicted-senescent cells, we generated receiver operating characteristic (ROC) curves for each of the markers. The ROC curve compares the true-positive rate versus the false-positive rate at increasing thresholds of detection, where high thresholds maximize true positives and low thresholds minimize false negatives. In this case, we classified cells as true positives if they remained CDK2$^{low}$ throughout the duration of the movie, which we approximated to be the true senescent subpopulation. For each marker, we computed two ROC curves: the first was for all cells in the population while the second was for only slow-cycling and predicted-senescent cells. This analysis allowed us to compare the relative resolving power for each senescence biomarker to differentiate 1) predicted-senescent cells from all other cells and 2) predicted-senescent from slow-cycling cells that pass through long periods of quiescence. Unsurprisingly, predicted-senescent cells are more easily resolved from fast-cycling cells than they are from slow-cycling cells that occasionally re-enter the cell cycle (Fig. 4e). Importantly, SA-β-Gal had the lowest ability to separate predicted-senescent cells from slow-cycling cells according to the ROC analysis (area-under-the-curve (AUC) = 0.55), whereas p21 and Lamin B1 had the highest ability to detect predicted-senescent cells (AUC = 0.75–0.76), with the other markers falling in between.

Next, we pooled all single-cell traces and binned cells into seven groups based on how many hours they spent in the CDK2$^{low}$ state during live-cell imaging from 6−10 d after etoposide release (Supplementary Fig. 7). We then averaged the intensity of each marker in each bin and plotted the result as a heatmap (Fig. 5, left and middle). This analysis clearly displays the graded nature of the increase (or decrease for Lamin B1) in marker intensities as a

function of time spent withdrawn from the cell cycle in response to etoposide treatment.

To test whether these results are generalizable to other types of senescence induction, we performed a similar experiment using hydrogen peroxide to induce oxidative stress, a common treatment for inducing senescence[28]. Here, cells were released from a 2 h treatment of 100 μM hydrogen peroxide and then immediately imaged for 4 days before being fixed and stained for the same senescence biomarkers. Although the percentage of predicted-senescent cells was lower after peroxide release than after etoposide release, we found a similar graded pattern in the staining of each marker with respect to cell-cycle status (Fig. 5, right, Supplementary Fig. 5, and Supplementary Fig. 7). These results indicate that our findings can likely be extended to additional methods of senescence induction and are independent of the level of population heterogeneity.

### Multiplexing senescence markers increases the accuracy of detecting predicted-senescent cells

While all the markers we assessed exhibited a graded increase in expression levels with respect to cell-cycle withdrawal time, p21 and Lamin B1 demonstrated the highest discriminative power in our ROC analysis (Fig. 4e). We hypothesized that the reason for this could be that the levels of p21 and Lamin B1 rapidly revert to their baseline intensities following cell-cycle re-entry, causing slow-cycling cells to reset their signals periodically compared to predicted-senescent cells. To test this idea, we compared the mean signal of p21 and Lamin B1 in slow-cycling cells that were in the cell cycle on the final frame of the movie vs. those that slowly cycled during the movie but were out of the cell cycle at the final frame of the movie. This allowed us to decouple the effects of current cell-cycle status from past proliferative history on the intensity of senescence biomarker staining. p21 levels fell immediately upon cell-cycle re-entry, consistent with prior time-lapse imaging of endogenously tagged p21[29], and Lamin B1 levels rose

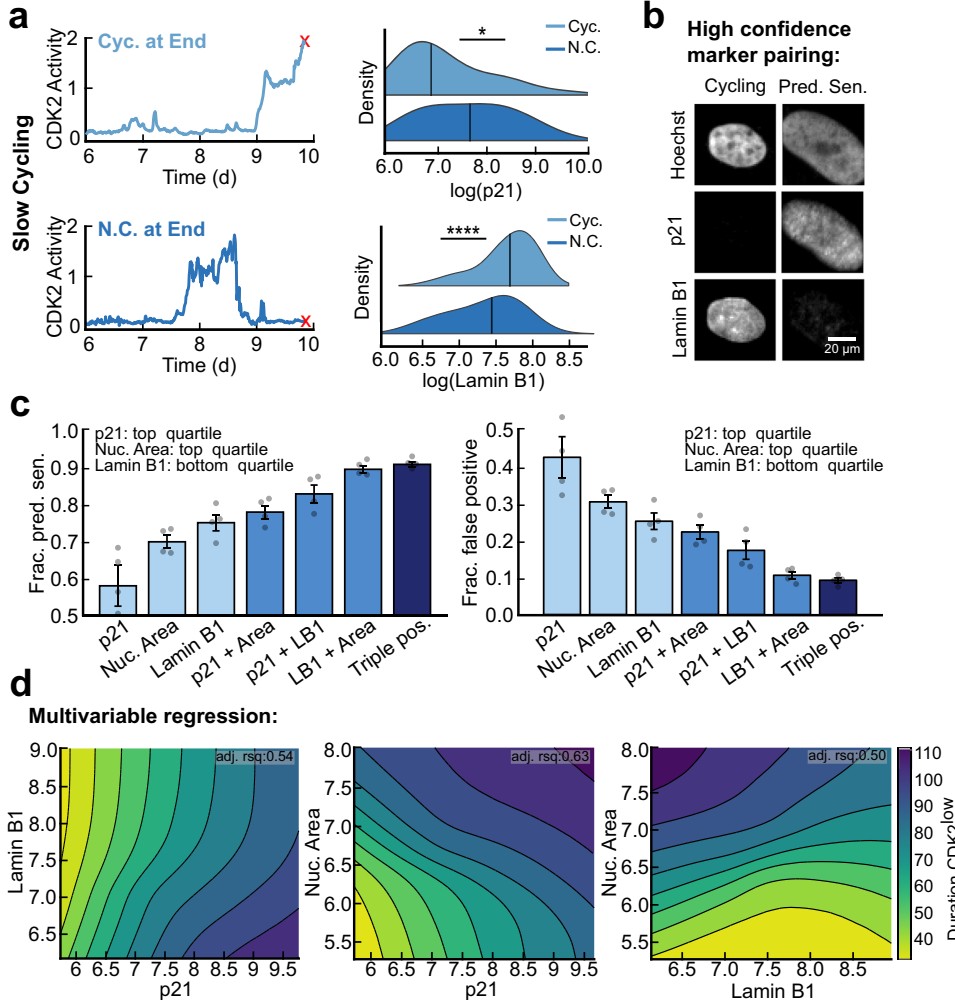

**Fig. 6 | p21 and Lamin B1 are short-memory markers that resolve predicted senescent cells. a** Single-cell traces from Fig. 4 were split into slow-cycling cells that completed a cell cycle during the 4 days of imaging but were CDK2[low] on the final frame of the movie (N.C. at End) vs. those that were in the cell cycle on the final frame of the movie (Cyc. at End). At the end of the movie, cells were stained for p21 and Lamin B1 and the distributions of p21 and Lamin B1 were plotted for the two groups. **b** MCF10A cells were co-stained for Hoechst, p21, and Lamin B1 after being imaged from 6-10 d after release from a 24 h treatment of 10 µM etoposide. **c** The fraction of predicted-senescent cells (true positives) and corresponding false positive rates were computed for every combination of markers using a binary cutoff at the top (p21, nuclear area) or bottom (Lamin B1) quartile of signal intensity. **d** Topological projections of local polynomial regression modeling of the data in **b** for each pair of senescence biomarkers with respect to the duration CDK2[low] during live-cell imaging.

immediately upon cell-cycle re-entry (Fig. 6a). Thus, p21 and Lamin B1 quickly "forget" the duration of cell-cycle withdrawal upon cell-cycle re-entry. This result supports the hypothesis that p21 and Lamin B1 can resolve predicted-senescent cells from slow-cycling cells because high levels of p21 and low levels of Lamin B1 are only achieved from long, continuous durations of cell-cycle withdrawal.

Next, we tested whether multiplexing the two markers with the highest AUC in Fig. 4e could increase our ability to identify cells that never cycled throughout the duration of imaging, which we designate as the predicted-senescent cells. After live-cell imaging from 6−10 d following 10 µM etoposide release, we co-stained cells for p21 and Lamin B1 (Fig. 6b). We also computed each cell's nuclear area, which we obtain for free from the time-lapse analysis (Fig. 6b). In order to have enough cells to assess the marker triple-plex, we applied generous thresholds for each marker by taking the top quartile of signal for p21 and nuclear area and the bottom quartile of signal for Lamin B1. With these cutoffs, p21 alone, nuclear area alone, and Lamin B1 alone correctly identified predicted-senescent cells 57%, 70%, and 75% of the time, respectively (Fig. 6c, left). The reason p21 performs worse here than Lamin B1 and nuclear area is because this analysis uses a single threshold, the top or bottom quartile, whereas the ROC analysis in

Fig. 4e scans all possible thresholds. Combining p21 and Lamin B1 raised the percentage of cells correctly predicted to 83%, and adding in nuclear area raised this value to 91% (Fig. 6c, left). The false-positive rate (flagging cells as senescent when they had actually cycled during the movie) followed the inverse trend (Fig. 6c, right).

While these data show that combining multiple markers in single cells increases the correct identification of predicted-senescent cells, the marker combinations and signal intensities at which the highest proportion of predicted-senescent cells could be captured remained unclear. To determine this, we performed several local polynomial regression analyses for each combination of marker pairings against the duration spent CDK2[low] throughout filming. This allowed us to generate several 3D surface projections whose topologies represent the proliferative likelihood of cells at different intensities of staining for each marker (Fig. 6d). From this, we found that as the intensities of any pair of markers rose (or fell for Lamin B1), the duration spent CDK2[low] increased proportionally, supporting our previous observation that senescence biomarker intensity at a snapshot encodes the proliferative history of single cells (Fig. 4d and Fig. 6d).

Lastly, we investigated whether the source of heterogeneity in marker staining could be associated with different levels of DNA

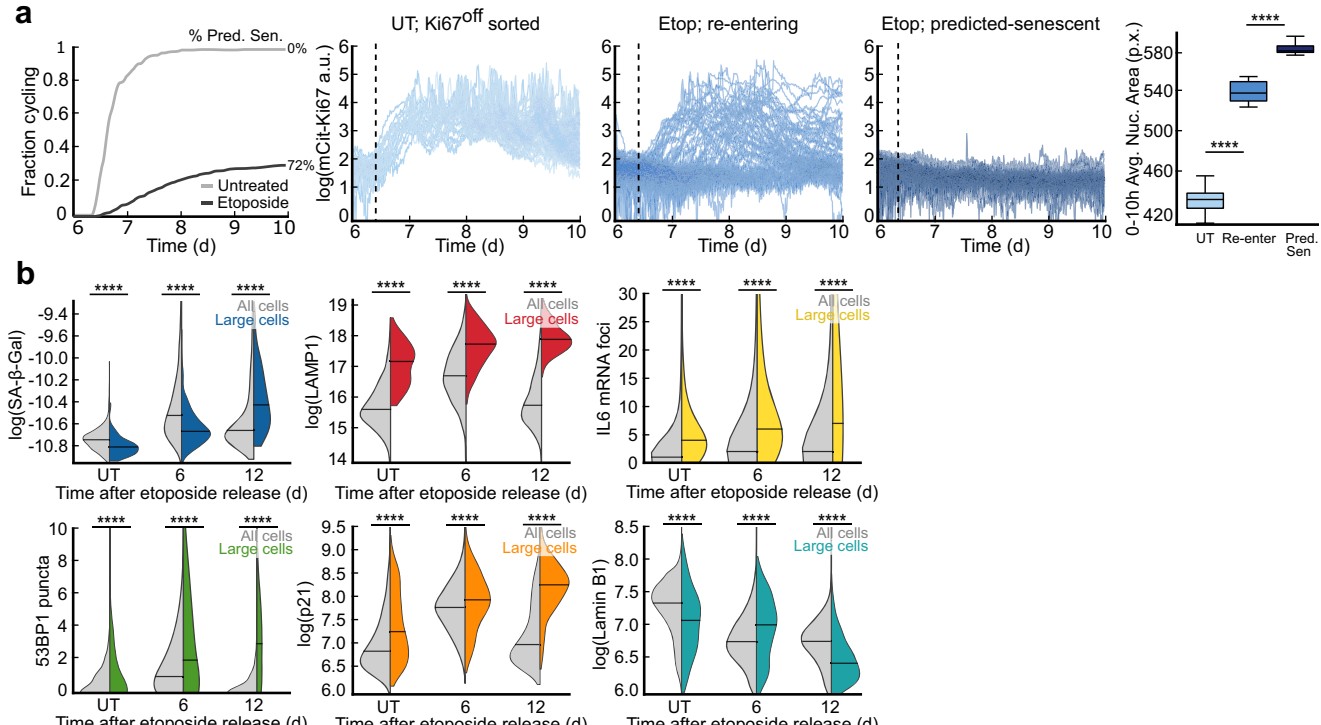

**Fig. 7 | The largest cells in the population go on to accumulate features of senescent cells. a** The data as in Fig. 1d. Cumulative distribution function (CDF) curves represent the fraction of cells that re-enter the cell cycle during the movie; the remainder of cells are defined as predicted-senescent (72%). Cells were clustered based on their relative mCitrine-Ki67 signals after sorting: untreated, re-entering, and predicted-senescent. The mean nuclear area within the first 10 h of filming was plotted for each category. UT, untreated. **b** SA-β-Gal, LAMP1, IL6 mRNA, 53BP1, p21, and Lamin B1 signal were measured at 6 d and 12 d after release from a 24 h treatment with 10 μM etoposide. Data are plotted for all cells (gray) vs. large cells (colors), which are defined as those with cytoplasmic areas >95th percentile of UT cells.

damage, which in turn may directly influence the extent of p21 induction. Although our ROC analysis did not reveal a large distinction between slow-cycling and predicted-senescent cells with 53BP1 staining, we suspected that this could be due to the discrete nature of this marker (i.e. cells can only have an integer number of foci). Despite these limitations, we multiplexed 53BP1 with p21 and nuclear area after a 6 d release from a 24 h treatment of 10 μM etoposide. Indeed, binning cells by the number of 53BP1 foci revealed that the number of 53BP1 foci scales with both p21 (Supplementary Fig. 8a) and nuclear area (Supplementary Fig. 8b), suggesting that heterogeneity in DNA damage gives rise to a gradient in the level of these two biomarkers. This is consistent with the fact that DNA damage activates p53, which upregulates p21[30,31], and with the notion that DNA damage can cause cells to slip from G2 to G1 phase without mitosis, creating large 4 N DNA content cells[32].

### The largest cells in the population go on to accumulate features of senescent cells

It was recently reported that increased cell size, one of the first identified and most common markers of senescence[33], is causal for senescence and the levels of senescence markers[34,35]. In this model, imbalanced scaling of the proteome causes some proteins to super-scale with size (e.g. SA-β-Gal and LAMP1) while other proteins subscale with size (e.g. Lamin B1)[34,36]. To place our findings in context of these observations, we returned to our dataset from Fig. 1d where untreated and etoposide-released mCitrine-Ki67 MCF10A cells were sorted to be Ki67off and replated for timelapse imaging the following day. Given that 28% of cells re-entered the cell cycle in this experiment, we classified cells as "re-entering" (etoposide-released and re-entered the cell cycle before the movie ended), or predicted-senescent (etoposide-released and remained Ki67off throughout the movie). Untreated cells that were

Ki67off at the time of sorting served as a control. We compared the cells' mean nuclear areas, a readout of cell size[37,38], in the first 10 h of filming when every cell was still Ki67off (Fig. 7a). Predicted-senescent cells had a significantly larger nuclear area than re-entering cells, which were significantly larger than untreated cycling cells in the 10 h window preceding escape from the Ki67off state (Fig. 7a). This result supports the notion that abnormal increases cell size are associated with reduced cell-cycle re-entry.

We next tested whether increased cell size was linked to a higher intensity of staining of canonical senescence markers over time. To test this, we measured SA-β-Gal, LAMP1, IL6 by mRNA fluorescence in situ hybridization (FISH), 53BP1, p21, and Lamin B1 at 6 d and 12 d following a 24 h treatment of 10 μM etoposide and compared the distributions for all cells versus the largest cells in the population (Fig. 7b). For the first 5 markers, the entire population rose and fell along the duration of recovery, since fast-cycling cells eventually outgrew predicted-senescent cells. However, within the subset of large cells, defined as those with cytoplasmic areas greater than the 95th percentile of untreated cells, the intensities of the markers continuously increased over time, while Lamin B1 followed the opposite trend (Fig. 7b). Thus, the largest cells following etoposide release accumulate a canonical senescent phenotype over time, while the remainder of the population likely re-enters the cell cycle and contributes to population regrowth.

## Discussion

Measuring senescence with either a single marker or at a single point in time can lead to incorrect conclusions about the biology and dynamics of senescent cells. Consistent with this notion, previous studies have reported that in certain contexts, cells can escape from senescence to resume proliferation in the future[2,39]. However, our time-resolved

analysis of single cells induced to senescence suggests that this regrowth phenotype stems from cells that were never truly senescent. Instead, quiescent cells that retain their capacity to proliferate can outcompete senescent cells over time in a heterogenous population. These cells may be misidentified as senescent, as they can express canonical senescence biomarkers, such as SA-β-Gal, which we have found to mark extended periods of cell-cycle withdrawal rather than senescence per se. In addition, our approach of matching each cell's stain to its cell-cycle history can explain marker combinations that do not "make sense" with traditional marker interpretation, such as the existence of SA-β-Gal[pos]/phospho-Rb[high] cells. Our results do not contradict previous literature that SA-β-Gal is associated with senescent cells, but rather demonstrate that temporary stress-induced cell-cycle withdrawal also induces SA-β-Gal expression, making it an imperfect marker of irreversible cell-cycle arrest. Despite these findings, SA-β-Gal is a convenient molecular marker due to its ease of use and can still provide valuable insights into the biology of long-term stress-induced cell-cycle exit.

To better understand the biological processes underlying SA-β-Gal staining, we examined the relationship between SA-β-Gal expression, lysosomal abundance, and autophagic flux. While some studies have reported autophagy to be suppressed during irreversible arrest[40], others have suggested that general autophagy is upregulated to enhance cell survival and inhibit proliferation during senescence[41]. Our results indicate that increased SA-β-Gal staining is associated with increased lysosomal mass and increased autophagic flux (Fig. 3h–i), consistent with previous studies that have shown autophagy to be activated in senescent cells[23,42].

We measured the intensities of additional senescence biomarkers following etoposide release and found that SA-β-Gal staining was not the only marker whose intensity was linked to a history of reduced proliferation. Indeed, increased cell size, LAMP1 staining, IL8, 53BP1 foci number, p21, and Lamin B1 were also strongly correlated with increasing time spent withdrawn from the cell cycle (Fig. 4). Of these, p21 and Lamin B1 had the highest resolving power for identifying predicted-senescent cells compared to slow-cycling cells because their signal intensities revert to baseline upon cell-cycle re-entry (Fig. 6a). However, even in combination, these markers cannot exclusively isolate predicted-senescent cells, calling into question the distinction between reversible and irreversible cell-cycle arrest.

This lack of clarity among cells at varying depths of cell-cycle withdrawal has been reported in recent literature. Most notably, Fujimaki et al. found that longer withdrawal from the cell cycle by serum starvation leads to a transcriptomic profile that increasingly resembles that of senescent cells, suggesting a continuum for cell-cycle exit[43]. Additionally, Stallaert et al. applied dimensionality reduction methods from hyperplexed imaging data of many cell cycle regulators after both quiescence and senescence induction and identified several graded arrest trajectories. They noted heterogeneity among nonproliferative cells across multiple established markers of arrest, where the cells with the highest staining intensity were the furthest from proliferating cells in high-dimensional space[9]. These observations support our findings that stress-induced cell-cycle withdrawal is graded rather than binary, where a cell's proliferative history is strongly linked to the intensities of senescence markers. Furthermore, because irreversible withdrawal from the cell cycle is not a prerequisite for cells to stain positive for senescence markers, the molecular features that separate quiescent from senescent cells remain unclear. However, it is still possible that there is an undiscovered marker of senescence that is entirely unique to irreversibly arrested cells that could accurately distinguish quiescent from senescent cells.

The graded relationship between marker intensity and cell-cycle withdrawal duration is perhaps unsurprising due to the interconnected nature of the signaling pathways that drive cellular senescence. Increased levels of DNA damage drive increased expression of p21[44,45],

which inhibits CDK2 activity to halt cell-cycle progression[14,29]. This blockade in proliferation causes a transcriptional decline in E2F target genes such as Lamin B1[46], which is also a substrate for selective autophagy in senescent cells[47]. Furthermore, increased lysosomal content (e.g., SA-β-Gal and LAMP1) and upregulation of the SASP (e.g., IL8 and IL6) are both linked to DNA-damage signaling[21,40,48] as well as imbalanced proteome scaling from increased cell size[34–36,49]. Thus, it is likely that none of the senescence markers are truly orthogonal to one another. We suggest that the molecular features associated with senescent cells are primarily caused by 1) DNA-damage-induced tumor suppressor signaling and 2) dysregulated cell size scaling, where the intensities of these signals reflect the level of stress experienced by the cell.

Our data suggest that stress-induced quiescence and senescence represent different degrees of cell-cycle withdrawal along a continuum rather than entirely distinct states, possibly explaining the field's struggle to identify black-and-white markers of senescence. If this continuum model is correct, it will be important to determine whether there is a point of no return for cell-cycle re-entry or if the probability of re-entry decreases steadily as cells progress along this continuum. This continuum model further suggests that it may be feasible to induce deeply quiescent cells to re-enter the cell cycle to promote tissue rejuvenation or to drive quiescent cancer cells towards senescence to prevent tumor recurrence. Importantly, our work demonstrates that snapshot data encodes dynamic information about the past, which can potentially be extrapolated to predict future cellular behavior. By devising strategies to retrieve such information, we may be able to uncover new mechanisms that govern cellular behavior in vivo. Such strategies could potentially aid the interpretation of pathology staining and enable clinicians to infer both past and future cellular behavior from fixed tissue samples.

## Methods

### Antibodies and reagents

Antibodies against Ki67 (ab15580) and LC3 II (ab192890) were purchased from Abcam and used at 1:2000 and 1:1000 dilutions. Antibodies against pRb (S807/811) D20B12 XP (8516), LAMP1 D2D11 XP (9091), p21 Waf1/Cip1 (12D1) (2947), Rb (4H1) (9309), and Lamin B1 (D9V6H) (13435) were purchased from CST and used at 1:500, 1:1000, 1:250, and 1:1000 dilutions, respectively. Antibodies against 53BP1 (612523), IL-8 (550419), and Waf1/Cip1/*CDKN1A* p21 (SX118) (sc-53870), which was used for Lamin B1 multiplexing, were purchased from BD and were all used at dilutions of 1:1000. Anti-LAMP1 (sc-20011), which was used for LC3 II multiplexing, was purchased from Santa Cruz Biotech and used at a 1:1000 dilution. All secondary antibodies, Goat anti-Mouse IgG (H + L) Cross-Adsorbed Secondary Antibody, Cyanine3 (A10521), Goat anti-Rabbit IgG (H + L) Cross-Adsorbed Secondary Antibody, Cyanine3 (A10520), Goat anti-Mouse IgG (H + L) Highly Cross-Adsorbed Secondary Antibody, Alexa Fluor 647 (A-21236), Goat anti-Rabbit IgG (H + L) Highly Cross-Adsorbed Secondary Antibody, Alexa Fluor 647 (A-21245) were purchased from Thermo Scientific and used at 1:1000 dilutions. IL6 FISH mRNA probe set (VA6–12712-VC) was purchased from Thermo Scientific. CF 488A succinimidyl ester (SCJ4600018) was purchased from Sigma and used at a 1:10,000 dilution. Hoechst 33342 was purchased from Thermo Scientific (H3570) and used at a 1:10,000 dilution. The colorimetric Senescence-Associated β-Gal Staining Kit was purchased from CST (9860). The fluorescent CellEvent Senescence Green Detection Kit (C10850) was purchased from Thermo Scientific. The ViewRNA ISH Cell Assay Kit was purchased from Thermo Scientific (QVC0001). Etoposide (E1383), Hydrogen Peroxide (108600) Chloroquine (AAJ6445914), and Brefeldin A (B7651) were purchased from Sigma. Palbociclib (S1116) and Trametinib (S2673) were purchased from Selleckchem.

## Cell lines and culture media

MCF10A (ATCC CRL−10317) cells were obtained from ATCC and grown in DMEM/F12 supplemented with 5% horse serum, 20 ng/ml EGF, 10 µg/ml insulin, 0.5 µg/ml hydrocortisone, 100 ng/ml cholera toxin, and 100 µg/mL of penicillin and streptomycin. MCF10A serum starvation media consisted of DMEM/F12, 0.5 µg/ml hydrocortisone, 100 ng/ml cholera toxin, and 100 µg/mL of penicillin and streptomycin. During live-cell imaging, phenol red-free full growth media was used. RPE-hTERT (ATCC CRL-4000) were obtained from ATCC and grown in DMEM/F12 supplemented with 10% FBS, 1x Glutamax, and 100 µg/mL of penicillin and streptomycin. MCF7 (ATCC HTB-22) were obtained from ATCC and grown in RPMI supplemented with 10% FBS, 1x Glutamax, and 100 µg/mL of penicillin and streptomycin. WI38-hTERT cells were obtained from the Campisi lab and grown in DMEM supplemented with 10% FBS and 100 µg/mL of penicillin and streptomycin. All cell lines were grown in a humidified incubator at 5% $CO_2$ and 37 °C.

## Drug treatments

MCF10A cells were plated at 100,000 cells per well in a plastic 6 well culture plate before being treated with 10 µM etoposide the following day for 24 h and then washed twice with PBS before being returned to full-growth media. The cells were maintained in culture throughout the duration of drug recovery with media refreshes every 3 d. 24 h prior to imaging, the etoposide-released cells were trypsinized and replated onto a collagen coated (1:50 dilution in water) (Advanced BioMatrix, No. 5005) 96-well glass-bottom plate (Cellvis Cat. No. P96−1.5H-N) at 1500 cells per well for live-cell imaging and 3000 cells per well for immunofluorescence. Similarly, to induce cells to senescence with hydrogen peroxide, cells were plated at 1500 cells per well in a 96-well glass-bottom plate and treated the following day with 100 µM $H_2O_2$ for 2 h in OptiMEM before being washed twice and returned to full growth media. To induce quiescence, 1500 cells per well were plated directly onto a collagen coated 96-well glass-bottom plate and treated the next day with 3 µM Palbociclib, 100 nM Trametinib, or serum-free media for up to 12 days. Contact-inhibited cells were plated at 10,000 cells per well in full-growth media in a 96-well glass-bottom plate and cultured for up to 12 days. Media was refreshed on all the conditions every 3 days. To perturb autophagy, MCF10A cells were treated with 50 µM chloroquine 3 h prior to fixing and staining. To visualize IL8, we blocked secretion for 6 h with 5 µg/mL Brefeldin A before fixing and staining.

## Flow cytometry

MCF10A cells endogenously tagged with mCitrine-Ki67 and expressing H2B-mTurquoise and DHB-mCherry were trypsinized and resuspended in PBS + 1% FBS + 100 µg/mL of penicillin and streptomycin after a 5 d recovery from a 24 h treatment with 10 µM etoposide or 10 Gy of ionizing radiation. Unlabeled wild-type cells were used to gate Ki67$^{off}$ cells, which resulted in 25% of etoposide-treated cells and 10% of IR-treated cells being sorted and replated directly onto a collagen-coated (1:50 dilution in water) (Advanced BioMatrix, No. 5005) 96-well glass-bottom plate (Cellvis Cat. No. P96−1.5H-N) for live-cell imaging that started the following day. As a control, the bottom 7.7% of untreated cells were also sorted and plated. For measuring SA-β-Gal in spontaneously quiescent cells, the bottom 1% of mCitrine-Ki67 was sorted and replated as described above for live-cell imaging that started 48 h later.

## Immunofluorescence

MCF10A cells were treated with 10 µM etoposide for 24 h, washed, and allowed to recover before being seeded onto a collagen coated (1:50 dilution in water) (Advanced BioMatrix, No. 5005) 96-well glass-bottom plate (Cellvis Cat. No. P96-1.5H-N) 24 h prior to fixation for 15 minutes with 4% PFA in PBS. Cells were permeabilized at room temperature with 0.1% TritonX for 15 minutes and blocked with 3% Bovine Serum Albumin (BSA) for 1 h. Primary antibodies were incubated overnight in 3% BSA at 4 °C and secondary antibodies were incubated for 1-2 h in 3% BSA at room temperature. Nuclei were labelled with Hoechst at 1:10,000 in PBS at room temperature for 15 min. Cytoplasms were labelled with succinimidyl ester 488 at 1:10,000 in PBS at room temperature for 30 minutes. Two 100 µL per well PBS washes were performed between each described step. All images were obtained using a 10 × 0.4 numerical aperture objective on a Nikon TiE microscope.

## Time-lapse microscopy

MCF10A cells were plated 24 h prior to imaging and full-growth media was replaced with phenol red-free full-growth media. Images were taken for each fluorescent channel every 12 minutes at two sites per well that were spaced 2 mm apart. Total exposure across all fluorescent channels was kept below 800 ms. Cells were imaged in a humidified, 37 °C chamber at 5% $CO_2$. All images were obtained using a 10 × 0.4 numerical aperture objective on a Nikon TiE microscope.

## Image processing

Image processing and cell tracking were performed using ImageJ 1.52e and MATLAB Mathworks 2017a as previously described[13]. Phospho-Rb was separated into high and low modes by using the saddle-point in the data as the cutoff (Supplementary Fig. 1a, b). mCitrine-Ki67$^{off}$ cells were classified as those less than the 95$^{th}$ percentile of the median nuclear signal in wild-type cells. Quantification of 53BP1 puncta was determined using a previously described approach[27]. Nuclear signals (phospho-Rb, 53BP1, p21, and Lamin B1) were quantified from a nuclear mask (median nuclear intensity), which was generated using Otsu's method on cells stained for Hoechst. Cytoplasmic signals (LAMP1, LC3, and IL8) were quantified from a cytoplasmic mask (median cytoplasmic intensity), which was generated using Otsu's method on cells stained for succinimidyl ester. Lysosomal masks for SA-β-Gal validation were generated from LAMP1 signal thresholding. The regionprops function in MATLAB was used to quantify the mean signal for each stain from these masks. Immunofluorescence and SA-β-Gal signals were linked back to live-cell imaging traces by nearest neighbor screening after jitter correction as described previously[13].

## SA-β-Gal quantification and validation

Compound immunofluorescence plus RGB images were obtained by mounting a LIDA light engine attachment to our Nikon TiE widefield microscope and exporting all stacked image channels from ND2 to TIFF via Nikon Elements Viewer. The SA-β-Gal stain for each cell is quantified by measuring the 5$^{th}$ percentile of the cytoplasmic red pixel intensity from pseudo-RGB images of the colorimetric stain (Fig. 2a). Cytoplasmic pixels were indexed from the binary mask generated with succinimidyl ester Alexa Fluor 488 as described above.

SA-β-Gal staining intensity is sensitive to the cell fixation method; 2% PFA and SA-β-Gal CST kit fixatives were compared (Supplementary Fig. 2a–c). Although the dynamic range of SA-β-Gal staining is larger for the kit fixative compared to 2% PFA, the kit fixative is less compatible with subsequent immunofluorescence staining (excluding LAMP1) (Supplementary Fig. 2c). The kit fixative was used for SA-β-Gal staining following all live-cell imaging experiments and LAMP1 immunofluorescence. 2% PFA was used for all other SA-β-Gal + immunofluorescence experiments.

To validate the SA-β-Gal quantification method, the upper and lower quartiles of population SA-β-Gal intensities were displayed through a binary cytoplasmic mask filter that was gated from the distribution of SA-β-Gal values after a 4d release from a 24 h pulse 10 µM etoposide (Supplementary Fig. 3a).

Immunofluorescence co-staining with SA-β-Gal is limited to the Cy5 channel due to strong bleed-through fluorescence in the GFP

channel and partial bleed-through into the Cy3 channel after staining with SA-β-Gal (kit fixative) (Supplementary Fig. 3b). Cy3 was only used for phospho-Rb (S807/811) co-staining since the bimodality of the phospho-Rb distribution is well maintained even after SA-β-Gal staining in the 2% PFA condition (Supplementary Fig. 2c).

To confirm that our 5th percentile quantification method for SA-β-Gal accurately represents the mean lysosomal signal, we used LAMP1 immunofluorescence to generate a lysosomal mask and calculated the mean SA-β-Gal signal in the lysosome (Supplementary Fig. 9a). Our 5th percentile method was found to be well-correlated with the mean lysosomal signal for every cell (Supplementary Fig. 9a, b).

Finally, to test the similarity between the colorimetric SA-β-Gal kit used throughout this work and the newer fluorescent SA-β-Gal method, we compared these two stains using both the 5th percentile and mean lysosomal methods. For both quantification methods, the colorimetric and the fluorescent SA-β-Gal signals were moderately correlated (Supplementary Fig. 9c, d).

### Receiver operating characteristic (ROC) analysis

We performed a ROC analysis by determining the false positive and true positive rate of detection for SA-β-Gal, LAMP1, cytoplasmic area, nuclear area, IL8, p21, and Lamin B1 by sliding the cutoff at every 10th percentile of intensity. The cutoff for 53BP1 was at increasing numbers of nuclear bodies (from 0 to 8+ foci). Our classification of true-positive senescent cells required that cells remained CDK2$^{low}$ throughout the duration of 4 days of live-cell imaging from day 6-10 after etoposide release.

### Statistical analyses and data quantification

The statistical tests used in our study were two-sample t-tests, with significance levels set at $*p < 0.05$, $**p < 0.01$, $***p < 0.001$, and $****p < 0.0001$ (Supplementary Data 1). Error bars in our figures indicate the standard error of the mean, which was calculated from multiple technical replicates (Supplementary Data 1). The technical replicates were obtained from data that was representative of at least two biological replicates. Our boxplots show the median and inter-quartile range, while the violin plots depict the median with black lines and the means with black dots. All instances of log refer to the natural log, which is the default in MATLAB.

### Reporting summary

Further information on research design is available in the Nature Portfolio Reporting Summary linked to this article.

## Data availability

The raw image data are available upon request to the corresponding author: Sabrina Spencer (sabrina.spencer@colorado.edu). No original code is reported in this paper. Source data are provided with this paper.

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

## Acknowledgements

We thank Judith Campisi for providing us with WI38-hTERT immortalized cells. We thank Theresa Nahreini, head of the Biochemistry Cell Culture Facility (BCCF), for her expertise and assistance with cell sorting. The Flow Cytometry Shared Core facility is supported by NIH grant S10OD021601. We thank Joe Dragavon, head of the Advanced Light Microscopy Core, for his insights on quantitative microscopy. We thank Steve Cappell and Iain Cheeseman for comments on the manuscript pre-submission. This work was supported by an NIH Training Grant T32 (5T32GM008759-19 (to H.M.A)), an NIH Training Grant T32 (5T32GM142607-21 (to B.F.)) and an NIH Director's New Innovator Award (1DP2CA238330-01 (to S.L.S)).

## Author contributions

H.M.A., B.F., and S.L.S. designed research; H.M.A. and B.F. conducted research; H.M.A. analyzed data; H.M.A. and S.L.S. conceived the project; and H.M.A. and S.L.S. wrote the paper; S.L.S. supervised the project.

## Competing interests

The authors declare no competing interests.
