## [Peer Review File · Nature Communications]

The intensities of canonical senescence biomarkers integrate the duration of cell-cycle withdrawalReviewers' Comments:

Reviewer #1:

Remarks to the Author:

In this manuscript, Ashraf et al. combine quantitative, time-lapse imaging with stainings of canonical senescence markers (SA- β -Gal staining, LAMP1 staining, cell size, DNA damage markers) to determine their predictive power to distinguish between reversibly-arrested and irreversibly-arrested cells. They conclude that individual "canonical" biomarkers of senescence generally reflect the duration of cell cycle arrest rather than its irreversibility and that the transition from reversible to irreversible arrest is accompanied by a gradual increase in biomarker strength rather than by an abrupt increase in signal. They suggest that these biomarkers should be used with caution and ideally combined to increase the ability to detect truly senescent cells.

Overall, this is a clearly written manuscript with straightforward conclusions. The systematic characterization of senescence biomarkers will be valuable for future senescence research. The authors' present well-designed experiments that make use of single cell reporters to support their conclusions. My concerns mostly stem from the fact that the authors have overlooked relevant literature — some of which has already demonstrated some of their conclusions using orthogonal methods. Though it is reassuring to see these findings affirmed using new methods, it detracts from the novelty of the authors' findings. Regarding the scientific quality, I believe that this manuscript will be suitable for publication after the concerns outlined below are addressed:

Major Comments:

1. The authors should confirm that the palbociclib, contact inhibition, MEK inhibition, and serum starvation conditions they show in Figure 3D are, in fact, reversible. They cite ref. 13 as evidence that these treatments are reversible, however ref. 13 shows this with a dose of palbociclib that is 3x lower than the dose used here. Also, ref. 13 uses a structurally distinct MEK inhibitor from that shown in this manuscript. If the authors wish to claim that high β -Gal staining is a function of cell cycle arrest duration and not irreversibility using these treatments, it is important that they show that these treatments are completely reversible under the conditions that they used.

2. Consistent with the data in Figure 5, work from Demidenko et al. 2008, Neurohr et al., Lanz, et al. 2022 and preprints from Wilson, et al. 2022, Manohar, et al. 2022, and Foy, et al. 2022 show that senescence following an induced cell cycle arrest is a function of cell size rather than duration of arrest. With regards to the authors' assertion that β -Gal intensity is a function of cell cycle arrest duration, Lanz, et al. and Wilson, et al. both show that constraining cell size during an 8-day arrest significantly lowers β -Gal staining. Consistent with this, the authors show that prolonged serum starvation (which precludes cell size increase) doesn't lead to a strong increase in β -Gal staining over time. Moreover, Lanz, et al. show that lysosome content (to which the authors attribute the increase in β -Gal) increases with cell size. If the authors wish to claim that "canonical senescence biomarkers integrate the duration of cell-cycle withdrawal," they should control for the effects of cell size increase during these treatments. This distinction is important because some cell lines naturally do not increase in size during a prolonged cell cycle arrest, and I suspect these cells would not stain for β -Gal (or maybe even other markers) even after a long arrest. Thus, addressing this will be useful for understanding how generalizable the authors' results are in other cell lines/systems.

3. This manuscript ignores several publications that have already demonstrated some of the results shown by the authors using orthogonal methods. Moreover, the authors make several assumptions/statements for which there are supporting data in the literature, but these data are frequently not cited. Some examples include:

- o On page 6, the authors make the (correct) assumption that nuclear area correlates with cell size, but provide no supporting literature reference.

o On page 4, the authors state that LAMP1 is “a membrane-embedded lysosomal protein” but provide no reference. They also do not explain what LC3II is.

o One of the main conclusions of this manuscript is that increased size is linked to developing senescence markers. Several recent papers and preprints (including those mentioned above in Major Comment #2) demonstrate that excess cell size is a cause of and not just a consequence of senescence.

o The authors’ “reasoned that increased β -Gal staining could simply be a readout of increased lysosomal content” because β -galactosidase is localized to lysosomes but cite no papers that demonstrate this. Increased lysosomal content has been observed in senescent cells since the 1970s, and Kurz, et al. 2000 demonstrates that SA- β -galactosidase simply reflects an increase in lysosomal content.

o Work from others (Yegorov, et al. 1998) has already demonstrated that high β -Gal levels do not necessarily diagnose senescence but rather non-proliferation.

Minor Comments:

1. The authors may want to reconsider plotting Figure 1B as a bar plot since the x-axis is not a discrete variable.

2. The labels on Figure 1C are a bit confusing. The cell cycle stage labels at the top seem like they should apply to both daughter cells because the labels on the bottom (time) do. The way it is shown (with the red G0 text off to the side) doesn’t make it clear that daughter cell #2 is not cycling. Because the “G0 (quiescence)” label is next to the “(Time since anaphase)” label, it appears that these are two separate categories being compared to each other.

3. For figures compiling multiple single cell measurements, the authors should include in the legend or in the methods section how many cells were analyzed for each condition.

Reviewer #2:

Remarks to the Author:

This manuscript examines the relationship between the intensities of some senescence biomarkers and the duration of cell cycle withdrawal. The problem of how to distinguish transient and permanent cell cycle arrest (if, in fact, cells ever truly exit the cell cycle permanently) is an important and unsolved problem. The paper contains useful data and could, potentially, be a useful contribution to the cell cycle field. However, it must first eliminate many instances of confusing language and overstated claims before being considered for publication.

Major Issues:

The largest issue with the paper is that the authors have claimed to examine permanent cell cycle arrest and senescence but have demonstrated neither. The use of the term “bona fide senescence” is particularly troubling because the authors only waited four days before claiming the cells were permanently arrested. What if they had waited three days? Or five days? The results in Figure 1E suggest that, had they waited longer, they would have seen more cells emerge from arrest. Similarly, the slope of the curve is still increasing in Figure 5A, and cells in Figure 5B continue to enter the cell cycle at the end of the movie. How many more cells re-entered 10 hours after the movie was completed? The study cannot make claims about permanent cell cycle withdrawal or bona fide

senescence without having sufficiently demonstrated these two phenotypes.

Moreover, it is important to acknowledge that permanent cell cycle withdrawal is a generally thought to be only one of many characteristics of the senescence phenotype, not its only feature. It would be more accurate to use a term such as "long-term cell cycle withdrawal" throughout the manuscript than use more extreme language such as permanent cell cycle arrest or senescence. Adjusting this use of terminology will help avoid confusing statements such as "these data support the notion that reversible and irreversible cell-cycle arrest exist on a continuum." This statement is not supported by data. It would be more accurate and helpful to say something like "either no truly irreversible arrest state exists, or we have yet to identify markers that sufficiently distinguish the irreversibly arrested state."

A second major issue is related to biomarkers. p16 and p21, two well-established markers of arrest and senescence, are mentioned in the introduction but not used in the paper. The manuscript would be much more compelling if p16 or p21 were shown to correlate with SA-beta-gal activity or if they could distinguish short-term and long-term arrest, one of the main goals of the paper. Of these two, p16 is most uniquely associated with senescence and should therefore be prioritized. Furthermore, in the discussion, the authors correctly remark that using a single senescence marker to identify senescence cells will lead to incorrect conclusions, and they mention in the introduction that "multiplexing multiple markers in single cells has been suggested as a new goal to identify senescent cells more accurately," citing papers from 2011 and 2015. In fact, recent studies have identified multiple markers of senescence including some examined in multiplex fashion (Cell Systems 13:230-240; Nature Aging 2:742-755; Nature Medicine 27:1941-1953). Incorporating knowledge from these papers into the present study could provide additional ways to find a distinction between short and long-term cell cycle withdrawal.

The paper demonstrates many of the problems with SA-beta-gal staining. This is important data for the senescence field to see because it will challenge the ways in which SA-beta-gal is conceptualized and used. In fact, the present manuscript refers to SA-beta-gal both as the "gold standard" while simultaneously demonstrating its shortcomings. However, the method used to quantify SA-beta-gal staining intensity is inferior to existing quantitative fluorescent kits. The authors should be cautious about making quantitative statements based on the colorimetric SA-beta-gal assay using a novel image analysis method that has not been benchmarked against more quantitative fluorescent versions of the assay. To make quantitative statements (e.g., showing proportionality to the duration of cell cycle withdrawal), the authors need to show a side-by-side comparison of their red-color-based quantification procedure with a fluorescence-based beta-gal activity assay, which should yield a near-linear increase with actual beta-gal activity.

More generally, the fact that there are a common set of molecular biomarkers used to mark both quiescence and senescence shows that we don't really know what proper biomarkers should be used to mark permanent cell cycle arrest. Therefore, it is misleading to use a set of imprecise biomarkers, demonstrate overlap between quiescence and senescence, and then claim the two states are therefore not distinct. If we have used markers of transient arrest to look for "permanently" arrested cells, are we really surprised that these markers are more intense under longer periods of arrest?

Minor Issues:

1. Referring to etoposide treatment in this context as chemotherapy is awkward. The authors are using etoposide as a DNA damaging agent. The drug becomes a chemotherapy when used on patients.
2. The abstract describes staining as being done immediately following treatment. This is not an accurate representation of the work.

3. The ratio of pRB to total RB is a better cell cycle indicator than pRB alone.
4. The paper missed some excellent opportunities to discuss the very similar and ongoing work by Dr. Guang Yao on understanding the depth of cell cycle arrest. Given the importance of this unsolved problem, it is critical to synthesize what is already known in the field so that the contributions of this study can be understood in relation to other scientific efforts.
5. For each figure that bins cells into specific time bins (e.g., 24-48 h), the authors should show the completed un-binned data in the supplement in order to give a sense of the overall trends in the timing of re-entry in real time.
6. The authors may consider adding a discussion and how and why SA-beta-gal has been used in practice and where it might be helpful or unhelpful.

Reviewer #3:

Remarks to the Author:

In this work, the authors used single-cell time-lapse imaging to study the associations of several senescence biomarkers (SA- β -Gal, LAMP1, cell size, and 53BP1) with individual quiescent cells and senescent cells (defined as those that did not reenter the cell cycle within 4 days after etoposide treatment). They found the intensities of these senescence biomarkers were graded rather than binary, primarily reflecting the duration of cell-cycle withdrawal. They proposed that quiescence and senescence fall on a continuum of cell-cycle re-entry likelihood rather than distinct cellular states. I found their findings quite interesting; they are consistent with and further expand some recent discoveries in the field (e.g., Ref 20, Fujimaki et al.).

It's well known that the senescence state is heterogeneous and signal-dependant. I'm curious how generalizable the main conclusion of this work is. Can the authors test the senescent state induced by a different signal (e.g., oncogenic or oxidative stress) and examine whether the senescence biomarker intensities also appear graded and reflect the duration of cell-cycle withdrawal? The result, if consistent, will further strengthen the current conclusion.

Minor points:

When slow-cycling cells were split into early vs. late escapers, no significant difference in the blueness levels between the two subpopulations was detected. The authors conclude that past proliferative history determines the final SA- β -Gal levels. It is not clear to me what evidence supports that the past proliferative history, not other factors, matters here. Can the authors please elaborate?

From Fig. 4E, the authors conclude that 53BP1 is the most enriched in senescent cells compared to quiescent cells, and thus the extent of DNA damage after etoposide release dictates the probability of cell-cycle re-entry. However, the difference of 53BP1 between senescent and quiescent cells was not more significant than those of other markers (Fig. 4B), and the AUC of 53BP1 to differentiate quiescence and senescence was actually smaller than those of two other markers (LAMP1 and Cyto.Area, Fig. 4D). The larger fold change of 53BP1 in Fig. 4E likely merely reflects its relatively small signal intensity in quiescent cells (Fig. 4B), rather than its unique biological significance.

In Fig. 5C, why the SA- β -Gal signal 6d after etoposide release was significantly smaller in large cells (presumably enriched for senescent cells) than the pop avg.?

When discussing the increased cell size as a senescence marker, why use nuclear areas in Fig 5A and 5B but the cytoplasmic area in Fig 5C and Fig 4? Shouldn't Fig 5A and 5B also use the cytoplasmic area (as an increase in the cytoplasm volume to DNA ratio better links to senescence)?

The authors conclude that increased autophagy may be a general feature of stress-induced cell-cycle exit and may control the probability of cell cycle re-entry. However, isn't this contradictory to the literature showing decreased autophagy is associated with senescence (e.g. doi.org/10.1038/nature16187, [10.1016/j.biocel.2017.09.005](https://doi.org/10.1016/j.biocel.2017.09.005))?

Overview

We thank the editor and the reviewers for their time and valuable suggestions related our manuscript at Nature Communications. All points raised by the reviewers have been incorporated into the revised version presented here and have significantly improved the manuscript. The major points that we addressed were:

- 1) Using less confusing language when discussing the phenotypes that we observed in this study.
- 2) Expanding the generalizability of the study by the addition of more molecular markers (we added four new markers), treatment conditions (we added oxidative stress as a senescence inducer), and analyses (we added a multiplexing analysis).
- 3) Citing additional relevant literature in the field to help contextualize the work, especially papers related to cell size, depth of quiescence, and senescence marker multiplexing.
- 4) Consideration of cell size as a driver of the phenotypes we observe.
- 5) Further validation for some of our newly described techniques and methods, particularly SA- β -Gal quantification.

We split each concern raised by the reviewers into three sections: comment, acknowledgement, and resolution. The comment is a copy-paste of the original comment; the acknowledgement is to ensure that we understand the context in which the reviewer is raising their concern; the resolution is how we dealt with the comment in the revised manuscript.

Reviewer #1

In this manuscript, Ashraf et al. combine quantitative, time-lapse imaging with stainings of canonical senescence markers (SA- β -Gal staining, LAMP1 staining, cell size, DNA damage markers) to determine their predictive power to distinguish between reversibly-arrested and irreversibly-arrested cells. They conclude that individual “canonical” biomarkers of senescence generally reflect the duration of cell cycle arrest rather than its irreversibility and that the transition from reversible to irreversible arrest is accompanied by a gradual increase in biomarker strength rather than by an abrupt increase in signal. They suggest that these biomarkers should be used with caution and ideally combined to increase the ability to detect truly senescent cells.

Overall, this is a clearly written manuscript with straightforward conclusions. The systematic characterization of senescence biomarkers will be valuable for future senescence research. The authors’ present well-designed experiments that make use of single cell reporters to support their conclusions. My concerns mostly stem from the fact that the authors have overlooked relevant literature — some of which has already demonstrated some of their conclusions using orthogonal methods. Though it is reassuring to see these findings affirmed using new methods, it detracts from the novelty of the authors’ findings. Regarding the scientific quality, I believe that this manuscript will be suitable for publication after the concerns outlined below are addressed:

Major Comments:

- 1) **Comment:** The authors should confirm that the Palbociclib, contact inhibition, MEK inhibition, and serum starvation conditions they show in Figure 3D are, in fact, reversible. They cite ref. 13 as evidence that these treatments are reversible, however ref. 13 shows this with a dose of palbociclib that is 3x lower than the dose used here. Also, ref. 13 uses a structurally distinct MEK inhibitor from that shown in this manuscript. If the authors wish to claim that high β -Gal staining is a function of cell cycle arrest duration and not irreversibility using these treatments, it is important that they show that these treatments are completely reversible under the conditions that they used.

Acknowledgement: This is a valid point, we did not show these treatments to be reversible in our original submission. In MCF10A cells, the MEK inhibitor PD-0325091 as well as lower doses of Palbociclib fail to induce stable, long-term arrest in all cells. Escape from proliferative arrest during the 2-week period of treatment would confound our results, which is why we used 100nM of the Mek inhibitor Trametinib and 3 μ M Palbociclib.

Resolution: To address this comment, we treated cells for 2 weeks with each of the four described treatments and then filmed the release using the CDK2 sensor. For contact inhibition, Mek inhibition, and serum starvation, 98%, 99%, and 98% of cells, respectively, re-entered the cell cycle within 5 days of release (**new Fig. 3E and new Fig. S4C**). However, 39% of 3 μ M Palbociclib-treated cells failed to cycle at all during the 5 day window of filming. These data suggest that a subset of Palbociclib-treated cells become senescent after long durations of quiescence induction. As a result, we removed language suggesting that the 3 μ M Palbociclib treatment is completely reversible. Fortunately, whether the cells are quiescent or senescent does not actually affect our overarching argument, since increasing durations of arrest (whether reversible or irreversible) yield a gradient in staining, which is the purpose of the figure.

- 2) **Comment:** Consistent with the data in Figure 5, work from Demidenko et al. 2008, Neurohr et al., Lanz, et al. 2022 and preprints from Wilson, et al. 2022, Manohar, et al. 2022, and Foy, et al. 2022 show that senescence following an induced cell cycle arrest is a function of cell size rather than duration of arrest. With regards to the authors' assertion that β -Gal intensity is a function of cell cycle arrest duration, Lanz, et al. and Wilson, et al. both show that constraining cell size during an 8-day arrest significantly lowers β -Gal staining. Consistent with this, the authors show that prolonged serum starvation (which precludes cell size increase) doesn't lead to a strong increase in β -Gal staining over time. Moreover, Lanz, et al. show that lysosome content (to which the authors attribute the increase in β -Gal) increases with cell size. If the authors wish to claim that "canonical senescence biomarkers integrate the duration of cell-cycle withdrawal," they should control for the effects of cell size increase during these treatments. This distinction is important because some cell lines naturally do not increase in size during a prolonged cell cycle arrest, and I suspect these cells would not stain for β -Gal (or maybe even other markers) even after a long arrest. Thus, addressing this will be useful for understanding how generalizable the authors' results are in other cell lines/systems.

Acknowledgement: Recent literature on the relationship between cell size and senescence is indeed important to consider. Lanz et al., 2022 show in Figure 5 of their manuscript that Palbociclib treatment causes cell size to increase as a function of treatment time, and Rapamycin co-treatment partially rescues the phenotype. Furthermore, controlling for size with this drug combination reduced the fraction of SA- β -Gal positive cells, suggesting that cell size controls the intensity of SA- β -Gal staining. However, closer inspection of Figure 5G-H of their manuscript shows that co-treatment of cells with Palbociclib and Rapamycin also reduces the abundance of p21 and p16, which are critical for maintaining cell-cycle arrest in senescent cells. As a result, their cells may well be cycling under Palbociclib/Rapamycin conditions, which may also explain the reduced SA- β -Gal.

Resolution: To address the concerns regarding contexts where cell size does not increase during long durations of cell-cycle withdrawal, such as serum starvation, we returned to our dataset in Figure 3E, where we plotted the intensities of SA- β -Gal as a function of long-term quiescence treatment duration. For each condition and timepoint, we plotted the distribution of the nuclear area (**new Fig. S4B**) and found that in every case but Palbociclib treatment, the cell size decreased with treatment duration, even though the cells became "bluer" over time. Thus, cells can become bluer even as they get smaller.

We do agree, however, that increased cell size is critical for the manifestation of a canonical senescent phenotype over time. As a result, we added more markers to our original list (**new Fig. 4**) to show that the signal intensity of each marker monotonically rises (or falls in the case of Lamin B1) in the largest cells over time (**new Fig. 7**). These findings support the model that increased cell size reinforces increased durations of proliferative arrest, which is integrated by the intensities of canonical senescence biomarkers. These data do not detract from our overarching argument, since we simply suggest that intensities of the biomarkers at a snapshot in time reflect how long the cells have been out of the cell cycle.

- 3) **Comment:** This manuscript ignores several publications that have already demonstrated some of the results shown by the authors using orthogonal methods. Moreover, the authors make several assumptions/statements for which there are supporting data in the literature, but these data are frequently not cited. Some examples include:

- o On page 6, the authors make the (correct) assumption that nuclear area correlates with cell size, but provide no supporting literature reference.
- O On page 4, the authors state that LAMP1 is “a membrane-embedded lysosomal protein” but provide no reference. They also do not explain what LC3II is.
- O One of the main conclusions of this manuscript is that increased size is linked to developing senescence markers. Several recent papers and preprints (including those mentioned above in Major Comment #2) demonstrate that excess cell size is a cause of and not just a consequence of senescence.
- O The authors’ “reasoned that increased β -Gal staining could simply be a readout of increased lysosomal content” because β -galactosidase is localized to lysosomes but cite no papers that demonstrate this. Increased lysosomal content has been observed in senescent cells since the 1970s, and Kurz, et al. 2000 demonstrates that SA- β -galactosidase simply reflects an increase in lysosomal content.
- O Work from others (Yegorov, et al. 1998) has already demonstrated that high β -Gal levels do not necessarily diagnose senescence but rather non-proliferation.

Acknowledgement: We agree that we can improve on our citations of the relevant literature.

Resolution: To address these issues, we now cite and discuss several papers in the revised version of our manuscript.

- a) Relationship between nuclear area and cell size (cell volume) 10.1083/jcb.200710156 & <https://doi.org/10.1091/mbc.e06-10-0973>
- b) LAMP1: DOI: 10.1006/cimm.1996.0167; LC3II: DOI: 10.1080/15548627.2018.1474314 & DOI: 10.1007/978-1-59745-157-4_4
- c) Cell size & senescence: Lanz et al.: 10.1016/j.molcel.2022.07.017, Neurohr et al.: 10.1016/j.cell.2019.01.018. Zatulovskiy et al.: <https://doi.org/10.3389/fcell.2022.980721>,
- d) Beta-galactosidase: DOI: Yegorov et al. 10.1006/excr.1998.4169, Kurz et al. : 10.1242/jcs.113.20.3613

Minor Comments:

- 1) **Comment:** The authors may want to reconsider plotting Figure 1B as a bar plot since the x-axis is not a discrete variable.

Resolution: Figure 1B was already plotted as a bar plot. Perhaps the reviewer was referring to a different subfigure, though it is unclear which one. We are happy to consider this request if the reviewer can clarify this point.

- 2) **Comment:** The labels on Figure 1C are a bit confusing. The cell cycle stage labels at the top seem like they should apply to both daughter cells because the labels on the bottom (time) do. The way it is shown (with the red G0 text off to the side) doesn’t make it clear that daughter cell #2 is not cycling. Because the “G0 (quiescence)” label is next to the “(Time since anaphase)” label, it appears that these are two separate categories being compared to each other.

Resolution: We resolved this by adding a separate label for each cell to indicate which is proliferating and which is quiescent in Figure 1C.

- 3) **Comment:** For figures compiling multiple single cell measurements, the authors should include in the legend or in the methods section how many cells were analyzed for each condition.

Resolution: We agree that having cell numbers for our single cell measurements is important for the reader. We included this information in a supplementary table along with replicates and p-values. Due to the large number of single cell measurements, putting this information into the already long figure captions would make them overly crowded. If required by the editor, however, we can move these cell numbers to the figure captions.

Reviewer #2

This manuscript examines the relationship between the intensities of some senescence biomarkers and the duration of cell cycle withdrawal. The problem of how to distinguish transient and permanent cell cycle arrest (if, in fact, cells ever truly exit the cell cycle permanently) is an important and unsolved problem. The paper contains useful data and could, potentially, be a useful contribution to the cell cycle field. However, it must first eliminate many instances of confusing language and overstated claims before being considered for publication.

Major issues:

- 1) **Comment:** The largest issue with the paper is that the authors have claimed to examine permanent cell cycle arrest and senescence but have demonstrated neither. The use of the term “bona fide senescence” is particularly troubling because the authors only waited four days before claiming the cells were permanently arrested. What if they had waited three days? Or five days? The results in Figure 1E suggest that, had they waited longer, they would have seen more cells emerge from arrest. Similarly, the slope of the curve is still increasing in Figure 5A, and cells in Figure 5B continue to enter the cell cycle at the end of the movie. How many more cells re-entered 10 hours after the movie was completed? The study cannot make claims about permanent cell cycle withdrawal or bona fide senescence without having sufficiently demonstrated these two phenotypes.

Acknowledgement: We'd like to point out that the movie was taken from day 6 to day 10 after etoposide release, not day 0 to day 4 as the reviewer suggests.

Indeed, identifying truly senescent cells is a major challenge holding this field back. The issue with establishing an unequivocal definition for irreversible arrest is that it would require continuous imaging of a cell for the entire duration of its lifecycle (months to years), which is technically infeasible. In this sense, senescence is more of a theoretical state that cells “asymptote” towards, where we can approximate cells to be in a senescent state if they remain in a CDK2^{low} state from day 6 to day 10 in our movie. This is already a vast improvement over the way in which senescence is described in the literature, where every cell is simply assumed to be irreversibly arrested due to the population average for various senescence markers being upregulated after various treatments. In our original submission, we can say with confidence that the cells we classified as quiescent proliferate more frequently than those that we classified as senescent. Furthermore, because we argue that withdrawal from the cell cycle is graded rather than binary, it does not detract from our overarching conclusion if a subset of the cells that we originally classified as senescent re-enter in the future, since these cells are at least “more arrested” than their neighbors.

Resolution: To address the reviewer's comment, we adjusted the x-axis movie data labels to begin at day 6 and end at day 10 to eliminate the confusion that Reviewer 2 and Reviewer 3 experienced. We also moved away from the terms proliferating, quiescent, and bona fide senescent labels and opted for fast cycling, slow cycling, and predicted-senescent (defined as cells that were CDK2^{low} for the entire movie). This is more in line with our main argument that withdrawal from the cell cycle is graded rather than binary, which is reflected by the gradient in staining of multiple senescence biomarkers.

- 2) **Comment:** It is important to acknowledge that permanent cell cycle withdrawal is generally thought to be only one of many characteristics of the senescence phenotype, not its only feature. It would be more accurate to use a term such as “long-term cell cycle withdrawal” throughout the manuscript than use more extreme language such as permanent cell cycle arrest or senescence. Adjusting this use of terminology will help avoid confusing statements such as “these data support the notion that reversible and irreversible cell-cycle arrest exist on a continuum.” This statement is not supported by data. It would be more accurate and helpful to say something like “either no truly irreversible arrest state exists, or we have yet to identify markers that sufficiently distinguish the irreversibly arrested state.”

Acknowledgement: We assume the reviewer is suggesting that senescent cells should be defined by the presence of certain molecular features or markers (e.g., SASP, SAHF, DNA damage, beta-

galactosidase etc.), and not necessarily irreversible cell-cycle arrest alone. We generally agree with this notion but note that while many of these features are associated with the functional roles of senescent cells *in vivo*, they are nearly all dispensable for the cell-cycle arrest phenotype. For example, p16 induces senescence without a SASP (Coppe, 2011); senescence-associated-heterochromatin foci only form in certain cell lines (Kosar, 2011); GLB1 mRNA depletion prevents the expression of beta-galactosidase but not the induction of senescence (Lee, 2006); and cytostatic drug combinations that do not induce a DNA damage phenotype (which we have confirmed) still cause cells to become senescent (Ruscetti, 2018). Thus, irreversible cell-cycle arrest is the only feature of senescent cells that is indispensable, which is why long-term tracking of cell-cycle reporters forms the basis of our work.

Resolution: To address this critique, we added a **new Figure 6** in which we multiplex several markers together. We also adjusted the terminology as the reviewer suggested away from proliferating, quiescent, and senescent to fast cycling, slow cycling, and predicted-senescent (defined as cells that were CDK2^{low} for the entire movie). The cells that do not cycle throughout the duration of the movie are the closest to senescent that we can approximate. Rather than stating that “irreversible and reversible cell-cycle arrest are on a continuum”, we now argue that cell-cycle withdrawal is graded rather than binary, a phenomenon consistent with two recent papers (E.g., Fujimaki 2019 & Stallaert, 2022). Even if these cells are not truly irreversibly withdrawn from the cell cycle, they are certainly “more arrested” than their neighbors. This is supported by the fact that these non-cycling cells have a more pronounced “senescence phenotype” (based on senescence biomarker staining) compared to cells we classified as slow cycling. Lastly, we took the reviewer’s suggestion for a helpful caveat statement and added a sentence in the Discussion stating, “However, it is still possible that there is an undiscovered marker of senescence that is entirely unique to irreversibly arrested cells that could accurately distinguish quiescent from senescent cells.”

- 3) **Comment:** p16 and p21, two well-established markers of arrest and senescence, are mentioned in the introduction but not used in the paper. The manuscript would be much more compelling if p16 or p21 were shown to correlate with SA-beta-gal activity or if they could distinguish short-term and long-term arrest, one of the main goals of the paper. Of these two, p16 is most uniquely associated with senescence and should therefore be prioritized. Furthermore, in the discussion, the authors correctly remark that using a single senescence marker to identify senescence cells will lead to incorrect conclusions, and they mention in the introduction that “multiplexing multiple markers in single cells has been suggested as a new goal to identify senescent cells more accurately,” citing papers from 2011 and 2015. In fact, recent studies have identified multiple markers of senescence including some examined in multiplex fashion (Cell Systems 13:230-240 ; Nature Aging 2:742–755 ; Nature Medicine 27:1941-1953). Incorporating knowledge from these papers into the present study could provide additional ways to find a distinction between short and long-term cell cycle withdrawal.

Acknowledgement: It is a good suggestion to stain for p16 and p21, two widely used markers of senescence. This is because they impinge directly on the cell cycle and thus play critical roles for maintaining cell-cycle withdrawal.

Resolution: In the revised manuscript, we now stain for p21 (along with additional markers IL8, Lamin B1, and nuclear area), following senescence induction and live-cell imaging. We chose p21 over p16 because 1) we have been unable to identify a good p16 antibody that works for immunofluorescence, 2) p16 is deleted in MCF10A cells as well as many other cells grown in culture, and 3) p16 is implicated in senescence primarily for replicative aging rather than acute DNA damage exposure, which activates p53 and p21 (Rayess, 2012, He, 2005). Importantly, we found that p21 intensity is also graded and scales with time spent out of the cell cycle. We also found that p21 has one of the highest AUCs in the ROC analysis and then determined why: levels of p21 rapidly revert to their baseline intensities following cell-cycle re-entry, causing slow-cycling cells to reset their signals periodically compared to predicted-senescent cells. We also discuss in more detail insights gained from previous literature on senescence marker multiplexing in our Discussion section. Specifically, we related our work to recent work by Fujimaki et al., 2019 and Stallaert et al., 2022 regarding the depth of cell-cycle exit and marker multiplexing.

- 4) **Comment:** The present manuscript refers to SA-beta-gal both as the “gold standard” while simultaneously demonstrating its shortcomings. However, the method used to quantify SA-beta-gal

staining intensity is inferior to existing quantitative fluorescent kits. The authors should be cautious about making quantitative statements based on the colorimetric SA- β -Gal assay using a novel image analysis method that has not been benchmarked against more quantitative fluorescent versions of the assay. To make quantitative statements (e.g., showing proportionality to the duration of cell cycle withdrawal), the authors need to show a side-by-side comparison of their red-color-based quantification procedure with a fluorescence-based beta-gal activity assay, which should yield a near-linear increase with actual beta-gal activity.

Acknowledgement: We agree that the dynamic range of fluorescent signals is generally stronger than colorimetric signals. However, most papers still use the colorimetric SA- β -Gal because this is the original stain published in the Dimri et al. 1995 paper and is considered the gold-standard. No one has shown, to our knowledge, that the fluorescent SA- β -Gal is equivalent to or superior to the colorimetric SA- β -Gal. Showing this equivalency requires staining with both fluorescent and colorimetric SA- β -Gal kits in the same single cells, which requires the special imaging set up that we describe in our work. Showing this equivalency also requires accurate quantification of these two signals to test the correlation in the same single cells. Our lab is now well-equipped to do this, and we have taken the reviewer's suggestion to heart (see below).

Because no previous automated quantification method for measuring colorimetric SA- β -Gal exists, we can only benchmark our quantification method against manual quantification, the standard practice in the field. Our quantification method accurately captures increasing "levels of blueness" for increasing intensities of staining (**Fig. 2B**). In other words, cells with more SA- β -Gal signal are visually bluer compared to cells with less SA- β -Gal signal. This is a strong indication that our pipeline is functioning as expected.

Resolution: We addressed the reviewer's comment by 1) further validating our quantification method and 2) comparing colorimetric SA- β -Gal staining to fluorescent SA- β -Gal staining. To further validate the quantification method for SA- β -Gal, we generated lysosomal masks by co-staining with LAMP1 and quantified the average SA- β -Gal signal within the lysosome of each cell. As expected, our 5th percentile method strongly correlated with our lysosomal-mask SA- β -Gal signal in single cells (**new Fig. S8**), suggesting that our 5th percentile method is a good proxy for the "true" colorimetric SA- β -Gal signal.

Additionally, we co-stained and quantified both colorimetric and fluorescent SA- β -Gal using both the 5th percentile method and the lysosomal-mask method in the same single cells. In both cases, we saw a positive linear correlation in single cells (**new Fig. S8**).

- 5) **Comment:** The fact that there are a common set of molecular biomarkers used to mark both quiescence and senescence shows that we don't really know what proper biomarkers should be used to mark permanent cell cycle arrest. Therefore, it is misleading to use a set of imprecise biomarkers, demonstrate overlap between quiescence and senescence, and then claim the two states are therefore not distinct. If we have used markers of transient arrest to look for "permanently" arrested cells, are we really surprised that these markers are more intense under longer periods of arrest?

Acknowledgement: Indeed, the set of markers used in the original submission of the manuscript have been shown to stain positive in both quiescent cells and senescent cells. However, these are not inherently "transient" markers of arrest, since they are primarily implicated in senescence. Because slow-cycling cells also stain positive for senescence markers, cells do not have to be irreversibly arrested to show senescence-like characteristics. This supports the idea that permanent withdrawal from the cell cycle, rather than the staining of senescence biomarkers, should define senescent cells, since senescence biomarkers are imprecise. However, we acknowledge that the caveat of this definition is that it is challenging to determine the cutoff for irreversibility.

Resolution: We added additional senescence markers (p21, Lamin B1, IL8, nuclear area) to our original set and found the same graded trend for these markers as well. We now acknowledge in the Discussion that it is still possible that there is an unidentified marker of senescence that is entirely unique to

irreversibly arrested cells. This does not detract from our overarching conclusion, which is that the senescence markers we test have a graded rather than a bimodal/binary distribution and reflect the duration of cell-cycle withdrawal.

Minor Comments:

- 1) **Comment:** Referring to etoposide treatment in this context as chemotherapy is awkward. The authors are using etoposide as a DNA damaging agent. The drug becomes a chemotherapy when used on patients.

Resolution: We modified the text as requested.

- 2) **Comment:** The abstract describes staining as being done immediately following treatment. This is not an accurate representation of the work.

Resolution: We updated the abstract to better reflect the paper's experimental workflow and also changed the axis labels on plots from hours to the actual days following release from etoposide.

- 3) **Comment:** The ratio of pRB to total RB is a better cell cycle indicator than pRB alone.

Resolution: It is true that normalizing phospho-Rb to total Rb gives a bit better separation between the two modes, but we have never seen a difference in our classification of cells whether we normalize to total Rb or not. We have included a new panel where we normalized the phospho-Rb signal to total Rb in untreated and etoposide-released cells to show that the fraction of cells classified as pRb^{pos} was the same with and without normalization (**new Fig. S1D**).

- 4) **Comment:** The paper missed some excellent opportunities to discuss the very similar and ongoing work by Dr. Guang Yao on understanding the depth of cell cycle arrest. Given the importance of this unsolved problem, it is critical to synthesize what is already known in the field so that the contributions of this study can be understood in relation to other scientific efforts.

Resolution: Fujimaki et al, 2019 was indeed an inspiration for this work. While we cited Fujimaki et al. in our original submission, we now discuss this paper in more detail to describe how it relates to our work. We do the same for Stallaert et al. 2022.

- 5) **Comment:** For each figure that bins cells into specific time bins (e.g., 24-48 h), the authors should show the completed un-binned data in the supplement in order to give a sense of the overall trends in the timing of re-entry in real time.

Resolution: We added heatmaps of the single-cell traces to supplemental figure 1 for all the Ki67 sorting conditions (**new Fig. S1F**). We also added heatmaps (**new Fig. S4C**) for the data underlying Figures 4 and 5 as well as violin plots showing the distribution of the marker signals in each bin (**Fig. S6**).

- 6) **Comment:** The authors may consider adding a discussion and how and why SA-beta-gal has been used in practice and where it might be helpful or unhelpful.

Resolution: In our Discussion, we added a sentence about the usage of SA-beta-gal and how it fits into the context of our findings.

Reviewer #3

In this work, the authors used single-cell time-lapse imaging to study the associations of several senescence biomarkers (SA- β -Gal, LAMP1, cell size, and 53BP1) with individual quiescent cells and senescent cells (defined as those that did not reenter the cell cycle within 4 days after etoposide treatment). They found the intensities of these senescence biomarkers were graded rather than binary, primarily reflecting the duration of cell-cycle withdrawal. They proposed that quiescence and senescence fall on a continuum of cell-cycle re-entry likelihood rather than distinct cellular states. I found their findings quite interesting; they are consistent with and further expand some recent discoveries in the field (e.g., Ref 20, Fujimaki et al.).

Note: We'd like to point out that the movie was taken from day 6 to day 10 after etoposide release, not day 0 to day 4 as the reviewer suggests. We adjusted the x-axis movie data labels to begin at day 6 and end at day 10 to eliminate the confusion that Reviewer 2 and Reviewer 3 experienced.

Major Comments:

- 1) **Comment:** It's well known that the senescence state is heterogeneous and signal dependent. I'm curious how generalizable the main conclusion of this work is. Can the authors test the senescent state induced by a different signal (e.g., oncogenic or oxidative stress) and examine whether the senescence biomarker intensities also appear graded and reflect the duration of cell-cycle withdrawal? The result, if consistent, will further strengthen the current conclusion.

Acknowledgement: We agree that a second treatment would increase the strength of our conclusions.

Resolution: As suggested by the reviewer, we induced cells to senescence with oxidative stress and obtained similar trends in the data where the marker intensities reflect the duration of cell-cycle withdrawal. Additionally, we added more biomarkers to increase the generalizability of the work.

Minor Comments:

- 1) **Comment:** When slow-cycling cells were split into early vs. late escapers, no significant difference in the blueness levels between the two subpopulations was detected. The authors conclude that past proliferative history determines the final SA- β -Gal levels. It is not clear to me what evidence supports that the past proliferative history, not other factors, matters here. Can the authors please elaborate?

Acknowledgement: We agree that this wording may have been confusing. The purpose of this experiment was to determine the origin of the SA- β -Gal^{pos}/phospho-Rb^{high} cells, which we find to be those that recently re-entered the cell cycle. In this case, the SA- β -Gal signal is maintained even upon commitment to the cell cycle due to its slow decay, and may require several cell cycles before returning to baseline.

Resolution: We adjusted the language for clarity to focus on the origin of the SA- β -Gal^{pos}/phospho-Rb^{high} cells. The late and early escaping cells (which we re-termed as slow-cycling cells that are vs. are not in the cell cycle on the final frame of the movie) are not significantly different in blueness because they spend similar durations withdrawn from the cell cycle—this point is not emphasized in the manuscript here because the following figure focuses on this idea.

- 2) **Comment:** From Fig. 4E, the authors conclude that 53BP1 is the most enriched in senescent cells compared to quiescent cells, and thus the extent of DNA damage after etoposide release dictates the probability of cell-cycle re-entry. However, the difference of 53BP1 between senescent and quiescent cells was not more significant than those of other markers (Fig. 4B), and the AUC of 53BP1 to differentiate quiescence and senescence was actually smaller than those of two other markers (LAMP1 and Cyto. Area, Fig. 4D). The larger fold change of 53BP1 in Fig. 4E likely merely reflects its relatively small signal intensity in quiescent cells (Fig. 4B), rather than its unique biological significance.

Acknowledgement: After more thought, we agree that the fold change is perhaps artificially large for 53BP1, probably because the units are discrete (0, 1, 2, 3 foci). The reviewer is correct that 53BP1 is not quantitatively the strongest senescence marker as measured by the ROC curves.

Resolution: We added several more senescence markers to our study and found p21 and Lamin B1 to be significantly stronger in resolving slow-cycling quiescent from non-cycling predicted-senescent cells according to the ROC curves. We performed a multiplexed analysis with p21 and Lamin B1 (along with nuclear area) to show how they may be used to detect cells likely to be senescent. Furthermore, we now show that the extent of damage scales with both p21 and cell size.

- 3) **Comment:** In Fig. 5C, why the SA- β -Gal signal 6d after etoposide release was significantly smaller in large cells (presumably enriched for senescent cells) than the pop avg.?

Acknowledgement: This is a reproducible finding that we have puzzled over for some time and have yet to come to a fully satisfying answer.

Resolution: We decided to leave the data in the paper but do not discuss this specific aspect since we do not fully understand the forces driving this observation and since it falls out of the scope of the major thrust of our paper. Below we have included some unpublished text and figures from a much older version of our work, in case the reviewer is interested in our attempts to explain this phenomenon.

“Excessive cytoplasmic dilution causes a functional decline in the ability of cells to synthesize new RNA and protein (Neurohr, 2019) and this could explain the delay in the rise of SA- β -Gal in large cells over time. We therefore measured each cell’s ratio of cytoplasmic area to DNA content after etoposide release and found that non-cycling pRb^{low} cells at 6d were indeed more dilute than untreated cells (**G**). However, by 12d, the non-cycling subpopulation’s dilution level began to return to baseline, indicating cellular re-concentration in the non-cycling cells over time (**G**). To test whether cellular dilution plays a role in SA- β -Gal positivity, etoposide-released cells were separated by the top 10% and bottom 10% of SA- β -Gal signal and their cytoplasmic dilution metrics and average protein abundance as measured by succinimidyl ester staining were compared (**H**). While SA- β -Gal^{high} cells were as concentrated as untreated cells, SA- β -Gal^{low} cells were the most dilute in the population (**H**). Thus, heterogeneity in SA- β -Gal staining may stem not only from differences in proliferation rate but also from variations in cellular concentration (**I, top**) and protein expression, a previously reported feature associated with increased SA- β -Gal staining (Kurz, 2000). In context of Neurohr et al., 2019 these data support the notion that cellular dilution may play a causal role in initiating early features of senescence; however, with extended durations of recovery, we suggest that cellular re-concentration in the non-cycling subpopulation may be critical for driving late features of senescence. This could occur if cell size increases up to its theoretical limit, but the net growth rate of cellular biomass remains constant, causing cellular re-concentration over time (**I, bottom**).”

- 4) **Comment:** When discussing the increased cell size as a senescence marker, why use nuclear areas in Fig 5A and 5B but the cytoplasmic area in Fig 5C and Fig 4? Shouldn't Fig 5A and 5B also use the cytoplasmic area (as an increase in the cytoplasm volume to DNA ratio better links to senescence)?

Acknowledgement: We used nuclear area from that experiment since our time-lapse movies always contain a fluorescent H2B nuclear marker for cell tracking, so we get nuclear area for free. We do not use a cytoplasmic marker for live-cell imaging, so we do not have the cytoplasmic area during the movie. Since nuclear and cytoplasmic area are phenotypically well-correlated, we have used them interchangeably.

Resolution: To address this comment, we now include both a nuclear area and a cytoplasmic area measurement the end of our movies in **new Figures 4 and 5** and found the same trends for both with respect to cell-cycle withdrawal duration.

- 5) **Comment:** The authors conclude that increased autophagy may be a general feature of stress-induced cell-cycle exit and may control the probability of cell cycle re-entry. However, isn't this contradictory to the literature showing decreased autophagy is associated with senescence (e.g. doi.org/10.1038/nature16187, 10.1016/j.biocel.2017.09.005)?

Acknowledgement: We agree that these data are contradictory of some literature on the topic; however, they are also in line with much other literature on the role of autophagy in promoting senescence.

Resolution: We added a section to our Discussion about the roles of autophagy in both promoting and inhibiting senescence in context of our results.

Reviewers' Comments:

Reviewer #1:

Remarks to the Author:

Overall, the authors have shown a good effort to address my points. There is one major and few minor remaining points that can be satisfied by changes to the text.

Major Comments

Comment 1: The authors have adequately addressed this comment.

Comment 2: I appreciate the effort the authors have undertaken to resolve this issue. The authors' hypothesis that the cells in Lanz, et al. 2022's experiment involving the combination of rapamycin + palbociclib allows cells to resume cycling due to a lack of p21/p16 expression is not likely, particularly since both palbociclib and mTOR inhibition arrest cells in G1 independently.

While the authors quantification of nuclear size rules out a sole role of cell size dictating β -gal staining, the serum starvation condition also speaks against prolonged cell cycle withdrawal being the main predictor of this phenotype. Unless serum starved cells occasionally divide and therefore accumulate less β -galactosidase. The authors should therefore either acknowledge that prolonged cell cycle withdrawal does not in all cases lead to (strong) β -galactosidase accumulation or provide evidence that serum starved cells are not efficiently arrested (EdU incorporation).

Comment 3: The authors have mostly addressed this comment. The authors still fail to reference seminal early observations about lysosomal content in senescent cells though (Robbins, et al. 1970). Also the recommended reference to Kurz, et al. but for some reason shows up twice in their reference list.

Minor Comment 1:

I apologize if my initial comment was unclear—I meant that the authors may want to reconsider their decision to plot Fig 1B as a bar plot, rather than that they should plot it as a bar plot. The current solution with showing an overlaid scatter/bar blot is satisfactory.

Minor Comment 2:

The authors have adequately addressed this comment.

Minor Comment 3:

The authors have adequately addressed this comment.

Additional Minor Comments:

In Figure 3D-E, the legend indicates that the measurements are for mean SA- β -gal whereas the y-axis of the figure indicates that it is the median SA- β -gal. The authors should correct this disparity.

Reviewer #2:

Remarks to the Author:

The manuscript is greatly improved, and I commend the authors for their thoughtful acknowledgement of the critiques and responsive revisions.

After a two additional revisions to the text of the manuscript, it is fit for publication.

1. The statement, "Furthermore, there is no robust method for quantifying SA- β -Gal. Most studies

simply binarize the colorimetric stain by manually labelling cells either blue (senescent) or not blue (not senescent)."

In fact, quantification of SA-BG can be done with fluorescent kits available on the market and used in this study.

I recommend changing the text to, "Furthermore, most studies neglect to use quantitative methods for measuring SA- β -Gal, such as fluorescent readouts. Instead, they simply binarize the colorimetric stain by manually labelling cells either blue (senescent) or not blue (not senescent). Due to these limitations, studies often measure other senescence markers in parallel experiments."

2. The claim "Since no single senescence marker is unique to senescence, multiplexing multiple markers in single cells has been suggested as a new goal to identify senescent cells more accurately" is misleading because multiplexing multiple markers is not a goal, but has already been explored in multiple papers.

I suggest changing the text to "...multiple markers in single cells to identify senescent cells is beginning to be explored." This more accurate statement should cite at least one of the papers that has already contributed to this goal.

Reviewer #3:

Remarks to the Author:

The authors have answered all my main questions in the revision. I congratulate the authors on this interesting and thorough analysis.

-- Guang Yao (signed)

Overview

We thank the editor and the reviewers for their time and valuable suggestions related our manuscript at Nature Communications. All points raised by the reviewers have been incorporated into the revised version presented here and have further improved the manuscript. We split each remaining concern raised by the reviewers into two sections: comment and resolution. The comment is a copy-paste of the original comment and the resolution is how we dealt with the comment in the revised manuscript.

Reviewer #1 (Remarks to the Author):

Overall, the authors have shown a good effort to address my points. There is one major and few minor remaining points that can be satisfied by changes to the text.

Major Comments

Comment 1: The authors have adequately addressed this comment.

Comment 2: I appreciate the effort the authors have undertaken to resolve this issue. The authors' hypothesis that the cells in Lanz, et al. 2022's experiment involving the combination of rapamycin + palbociclib allows cells to resume cycling due to a lack of p21/p16 expression is not likely, particularly since both palbociclib and mTOR inhibition arrest cells in G1 independently.

Resolution: *Yes, the reviewer makes a good point that it is unlikely that cells in CDK4/6i + mTORi are cycling. We agree that increased cell size is a critical feature of senescence and that restraining cell size in the presence of senescence inducing treatments (e.g. using mTor inhibitors) can reduce the manifestation of many senescence phenotypes.*

While the authors quantification of nuclear size rules out a sole role of cell size dictating β -gal staining, the serum starvation condition also speaks against prolonged cell cycle withdrawal being the main predictor of this phenotype. Unless serum starved cells occasionally divide and therefore accumulate less β -galactosidase. The authors should therefore either acknowledge that prolonged cell cycle withdrawal does not in all cases lead to (strong) β -galactosidase accumulation or provide evidence that serum starved cells are not efficiently arrested (EdU incorporation).

Resolution: *Our serum-starved cells are indeed nearly fully arrested.*

We agree with the reviewer that prolonged cell-cycle arrest is not sufficient to produce strong SA- β -gal accumulation under all arrest conditions, given our serum starvation data. However, our quantification of nuclear size demonstrates that increased cell size is not the only factor dictating SA- β -gal accumulation, given that trametinib-treated cells become both bluer and smaller. These data suggest that both cell size and prolonged cell-cycle withdrawal are strongly associated with senescence but neither is solely responsible for SA- β -gal accumulation.

We have therefore revised the text to clearly state that prolonged cell cycle withdrawal does not lead to strong β -galactosidase accumulation in serum starved cells. While we do not know the

full reason behind this, we suggest that serum starved cells do not have strong SA- β -gal staining because they are arrested in a natural fashion akin to the way quiescent cells are arrested in the body. This condition lacks a clash between pro- and anti-proliferation signals and does not yield SA- β -gal accumulation as strongly as conditions where cells are arrested but maintain pro-growth signaling. Cells induced to senescence but cotreated with rapamycin may have low SA- β -gal for similar reasons, but this remains to be tested in future work.

Comment 3: The authors have mostly addressed this comment. The authors still fail to reference seminal early observations about lysosomal content in senescent cells though (Robbins, et al. 1970). Also the recommended reference to Kurz, et al. but for some reason shows up twice in their reference list.

Resolution: Thank you for pointing this out. We have updated the text to reference Robbins, et al. 1970 and fixed the duplicated citation for Kurz, et al.

Minor Comment 1:

I apologize if my initial comment was unclear—I meant that the authors may want to reconsider their decision to plot Fig 1B as a bar plot, rather than that they should plot it as a bar plot. The current solution with showing an overlaid scatter/bar blot is satisfactory.

Minor Comment 2:

The authors have adequately addressed this comment.

Minor Comment 3:

The authors have adequately addressed this comment.

Additional Minor Comments:

In Figure 3D-E, the legend indicates that the measurements are for mean SA- β -gal whereas the y-axis of the figure indicates that it is the median SA- β -gal. The authors should correct this disparity.

Resolution: This disparity has been corrected and the figure legend has been adjusted to "median".

Reviewer #2 (Remarks to the Author):

The manuscript is greatly improved, and I commend the authors for their thoughtful acknowledgement of the critiques and responsive revisions.

After a two additional revisions to the text of the manuscript, it is fit for publication.

1. The statement, "Furthermore, there is no robust method for quantifying SA- β -Gal. Most studies simply binarize the colorimetric stain by manually labelling cells either blue (senescent) or not blue (not senescent)."

In fact, quantification of SA-BG can be done with fluorescent kits available on the market and used in this study.

I recommend changing the text to, "Furthermore, most studies neglect to use quantitative methods for measuring SA- β -Gal, such as fluorescent readouts. Instead, they simply binarize the colorimetric stain by manually labelling cells either blue (senescent) or not blue (not senescent). Due to these limitations, studies often measure other senescence markers in parallel experiments."

Resolution: *We appreciate the reviewer's point that fluorescent kits are available for quantification of SA- β -Gal but are not widely used, and have changed the text to reflect this point as suggested, with updated wording.*

"Furthermore, the colorimetric nature of the standard SA- β -Gal stain makes it challenging to quantify. While more quantitative fluorescent senescence-detection kits now exist, most studies still classify cells by simply binarizing the classic colorimetric SA- β -Gal stain by manually labelling cells either blue (senescent) or not blue (not senescent). Due to these limitations, studies often measure additional senescence markers in separate parallel experiments..."

2. The claim "Since no single senescence marker is unique to senescence, multiplexing multiple markers in single cells has been suggested as a new goal to identify senescent cells more accurately" is misleading because multiplexing multiple markers is not a goal, but has already been explored in multiple papers.

I suggest changing the text to "...multiple markers in single cells to identify senescent cells is beginning to be explored." This more accurate statement should cite at least one of the papers that has already contributed to this goal.

Resolution: *We have changed the text to reflect that recent studies have started multiplexing markers and cite a recent paper that has contributed to this goal (Stallaert et al. 2022).*

Reviewer #3 (Remarks to the Author):

The authors have answered all my main questions in the revision. I congratulate the authors on this interesting and thorough analysis.

-- Guang Yao (signed)